EMBO
Molecular Medicine

# The mitochondrial calcium uniporter regulates breast cancer progression via HIF-1α

Anna Tosatto[1], Roberta Sommaggio[2], Carsten Kummerow[3], Robert B Bentham[4], Thomas S Blacker[4], Tunde Berecz[4], Michael R Duchen[4], Antonio Rosato[2,5], Ivan Bogeski[3], Gyorgy Szabadkai[1,4], Rosario Rizzuto[1,6,*] & Cristina Mammucari[1,**]

## Abstract

Triple-negative breast cancer (TNBC) represents the most aggressive breast tumor subtype. However, the molecular determinants responsible for the metastatic TNBC phenotype are only partially understood. We here show that expression of the mitochondrial calcium uniporter (MCU), the selective channel responsible for mitochondrial $Ca^{2+}$ uptake, correlates with tumor size and lymph node infiltration, suggesting that mitochondrial $Ca^{2+}$ uptake might be instrumental for tumor growth and metastatic formation. Accordingly, MCU downregulation hampered cell motility and invasiveness and reduced tumor growth, lymph node infiltration, and lung metastasis in TNBC xenografts. In MCU-silenced cells, production of mitochondrial reactive oxygen species (mROS) is blunted and expression of the hypoxia-inducible factor-1α (HIF-1α) is reduced, suggesting a signaling role for mROS and HIF-1α, downstream of mitochondrial $Ca^{2+}$. Finally, in breast cancer mRNA samples, a positive correlation of *MCU* expression with HIF-1α signaling route is present. Our results indicate that MCU plays a central role in TNBC growth and metastasis formation and suggest that mitochondrial $Ca^{2+}$ uptake is a potential novel therapeutic target for clinical intervention.

**Keywords** breast cancer; HIF-1α; metastasis; mitochondrial $Ca^{2+}$ uptake; reactive oxygen species

**Subject Categories** Cancer; Metabolism

## Introduction

Mitochondrial $Ca^{2+}$ uptake regulates cellular energetics by triggering ATP synthesis. At the same time, mitochondrial $Ca^{2+}$ acts as a key controller of both cell metabolism and fate. Indeed, a decrease in ATP production induces autophagy, while $Ca^{2+}$ overload causes organelle dysfunction and release of caspase cofactors (Rizzuto *et al*, 2012). Several pathological conditions, including tumor formation and progression, are directly related to mitochondrial dysfunctions, and reprogramming of mitochondrial metabolism is now considered as an emerging hallmark of cancer (Hanahan & Weinberg, 2011). Indeed, even in the presence of oxygen, cancer cells limit their energy supply largely to glycolysis, leading to the so-called aerobic glycolysis phenotype (Sciacovelli *et al*, 2014). Of note, the dependence on glycolytic fueling is further potentiated by hypoxia, a condition that characterizes most tumor microenvironments. In response to oxygen deprivation, the hypoxia-inducible factor-1α (HIF-1α) is stabilized and transcription of glucose transporters and glycolysis-related enzymes, which are HIF-1α target genes, is induced (Semenza, 2010). In addition, in specific settings, altered mitochondrial metabolism represents a primary trigger for cancer progression, as demonstrated by several hereditary tumors associated with mutations in key mitochondrial enzymes (Gottlieb & Tomlinson, 2005). Consistent with these observations, among the most aggressive human breast tumors, triple-negative breast cancers (TNBCs), a clinically heterogeneous category of breast tumors that lack expression of estrogen receptor, progesterone receptor, and human epidermal growth factor receptor 2 (HER2), show profound metabolic alterations with impaired mitochondrial oxidative metabolism (Elias, 2010; Owens *et al*, 2011). In these complex tumorigenic settings, mitochondrial reactive oxygen species (mROS), as by-products of mitochondrial respiratory chain electron flux, play a fundamental role (Roesch *et al*, 2013). mROS are essential molecules for intracellular communication, preserving cell homeostasis and triggering adaptation to stress (Wu, 2006; Sena & Chandel, 2012). Moreover, mROS have been defined as crucial molecular effectors for cancer progression, by eliciting both metabolic adaptations and *in vivo* metastasis formation (Tochhawng *et al*, 2013; Porporato *et al*, 2014; Cierlitza *et al*, 2015).

---

1 Department of Biomedical Sciences, University of Padua, Padua, Italy
2 Department of Surgery, Oncology and Gastroenterology, University of Padua, Padua, Italy
3 Department of Biophysics, Center for Integrative Physiology and Molecular Medicine (CIPMM), School of Medicine, Saarland University, Homburg, Germany
4 Department of Cell and Developmental Biology, Consortium for Mitochondrial Research, University College London, London, UK
5 Veneto Institute of Oncology IOV - IRCCS, Padua, Italy
6 CNR Institute of Neuroscience, National Council of Research, Padua, Italy
*Corresponding author: Tel.: +39 049 8273001; E-mail: rosario.rizzuto@unipd.it
**Corresponding author: Tel.: +39 049 8276481; E-mail: cristina.mammucari@unipd.it

The mitochondrial calcium uniporter (MCU), the channel responsible for mitochondrial $Ca^{2+}$ uptake, has been recently identified (Baughman *et al*, 2011; De Stefani *et al*, 2011). A number of proteins contribute to the channel complex (Raffaello *et al*, 2013; Sancak *et al*, 2013; Foskett & Philipson, 2015) and others regulate its activity (Perocchi *et al*, 2010; Plovanich *et al*, 2013; Patron *et al*, 2014), but little is known about the role of MCU-dependent mitochondrial $Ca^{2+}$ homeostasis in tumor progression. Recent evidence indicates that prostate and colon cancers overexpress an MCU-targeting microRNA that, by reducing mitochondrial $Ca^{2+}$ uptake, favors cancer cell resistance to apoptotic stimuli, thus increasing cell survival (Marchi *et al*, 2013). Moreover, constitutively elevated mitochondrial $Ca^{2+}$ influx triggers mROS generation and enhances the sensitivity of HeLa cells to ceramide-induced cell death (Mallilankaraman *et al*, 2012). However, a recent study reported a correlation between *MCU* overexpression and poor prognosis in breast cancer patients (Hall *et al*, 2014). Furthermore, in the MDA-MB-231 cell line, a TNBC model, caspase-independent cell death was potentiated by MCU silencing, suggesting that MCU overexpression may offer a survival advantage against some apoptotic pathway (Curry *et al*, 2013). Finally, the role of MCU in the control of breast cancer cell migration has been ascribed to a store-operated $Ca^{2+}$ entry-dependent mechanism (Tang *et al*, 2015).

Here, we show that *MCU* expression correlates with breast tumor size and lymph node infiltration. MCU silencing causes a significant decline in mitochondrial $[Ca^{2+}]$, metastatic cell motility, and matrix invasiveness. Most importantly, in MDA-MB-231 xenografts, deletion of *MCU* greatly reduces tumor growth and metastasis formation. In the absence of MCU, production of mROS is significantly lower, suggesting that mROS might play a crucial role in cell malignancy regulation by mitochondrial $Ca^{2+}$ uptake. Moreover, MCU silencing downregulates HIF-1α expression, thus impairing the transcription of HIF-1α-target genes involved in tumor progression. In agreement with HIF-1α being a major effector of MCU, rescue of HIF-1α expression restores migration of MCU-silenced TNBC cells. Finally, breast cancer dataset analysis confirms a strong correlation of *MCU* expression with HIF-1α signaling. In conclusion, our work points out MCU as a critical checkpoint of metastatic behavior, and thus a potential pharmacological target in aggressive cancers, such as TNBC.

# Results

## *MCU* expression correlates with breast tumor progression and cell migration

To decipher the role of mitochondrial $Ca^{2+}$ signaling in metastatic potential, we collected the mRNA levels of MCU and related proteins (MCUb, MICU1-3, and EMRE) from the TCGA breast cancer dataset (http://tcga-data.nci.nih.gov/docs/publications/brca_2012/) (Koboldt *et al*, 2012). Data analyses relative to tumor size and regional lymph node infiltration demonstrate a significant correlation of *MCU* and *MCUb* expression levels with breast cancer clinical stages (Fig 1A and B). In particular, while *MCU* expression increases with tumor progression, the expression of *MCUb*, the dominant-negative channel isoform, decreases. These data suggest that mitochondrial $Ca^{2+}$ uptake may increase with tumor size and infiltration. On the

other hand, no correlation of the expression of MCU regulators (*MICU1-3* and *EMRE*) with tumor size and lymph node infiltration was detected (Appendix Fig S1A and B), suggesting that either no control is exerted on MCU regulators or that post-translational modifications may be critical (Patron *et al*, 2014; Petrungaro *et al*, 2015).

These data indicate that increased mitochondrial $Ca^{2+}$ uptake may be instrumental for metastasis. We decided to verify this hypothesis in a specific breast tumor subset, that is, TNBC. Accordingly, three different human metastatic TNBC models were analyzed: BT-549, MDA-MB-468, and MDA-MB-231 cell lines. For each cell line, an agonist that evokes a robust cytosolic $Ca^{2+}$ transient was chosen (i.e., ATP for MDA-MB-231 and MDA-MB-468, histamine for BT-549 cells). In all three cell models, short-interfering RNA (siRNA)-mediated inhibition of MCU caused a significant decline in agonist-induced mitochondrial $Ca^{2+}$ uptake (Fig 1C–E).

In line with the consistent effect on mitochondrial $Ca^{2+}$ uptake, MCU silencing impaired cell motility, monitored by wound healing migration assay, in all TNBC lines tested (Fig 1F–H), while proliferation was largely unaffected (Fig 1I–K). The inhibitory effect of MCU silencing on MDA-MB-231 cell migration has been previously ascribed to the regulation of store-operated $Ca^{2+}$ entry (SOCE), although the mechanism remains unclear (Tang *et al*, 2015). To clarify whether the impairment of migration is specifically due to the reduction in mitochondrial $Ca^{2+}$ uptake, or rather to indirect effects of MCU silencing on global cellular $Ca^{2+}$ signaling, cytosolic $Ca^{2+}$ transients, SOCE, and ER $Ca^{2+}$ content were measured. MCU silencing caused a decrease of agonist-induced cytosolic $Ca^{2+}$ transients in BT-549 and MDA-MB-231 cell lines but not in MDA-MB-468 (Appendix Fig S2A), maybe reflecting a cell type-specific regulation of the inhibitory role that local high $[Ca^{2+}]$ microdomains play on Ins(1,4,5)P3R activity (Rizzuto *et al*, 2012). In contrast to what was previously reported (Tang *et al*, 2015), MCU silencing caused an increase in SOCE in MDA-MB-231 and MDA-MB-468 cell lines, in terms of both speed and maximal $[Ca^{2+}]$ entry and irrespective of the experimental protocol used to deplete $Ca^{2+}$ store (either CPA, ionomycin or Ins(1,4,5)P$_3$-coupled agonist) (Appendix Fig S2B–D). However, this effect was absent in BT-549 cells. Treatment with CPA or ionomycin in the absence of extracellular $Ca^{2+}$ demonstrated that MCU silencing does not affect intracellular $Ca^{2+}$ stores in all cell lines here tested (Appendix Fig S2B–D). Overall, these results indicate a cell line-dependent effect of MCU knockdown on the regulation of cytosolic $Ca^{2+}$ transients and SOCE in the different TNBC lines analyzed. Thus, the impairment in cell migration triggered by MCU silencing is most likely due to the specific reduction in mitochondrial $Ca^{2+}$ uptake that was consistently observed in the three cell lines, as opposed to the other aspects of global $Ca^{2+}$ homeostasis.

To complete the picture, overexpression of MCU triggered an increase in agonist-induced mitochondrial $Ca^{2+}$ uptake as expected (Appendix Fig S3A), and a decrease in cytosolic $[Ca^{2+}]$ transients in the three cell lines (Appendix Fig S3B), indicating that increased MCU levels can uncover the buffering role that mitochondria can exert on cytosolic $Ca^{2+}$ rises (De Stefani *et al*, 2011). MCU overexpression did not affect intracellular $Ca^{2+}$ stores, as demonstrated by CPA, ionomycin, or agonist treatments in $Ca^{2+}$-free media (Appendix Fig S3C–E). Finally, the effect caused by MCU overexpression on SOCE was only marginal (Appendix Fig S3C–E).

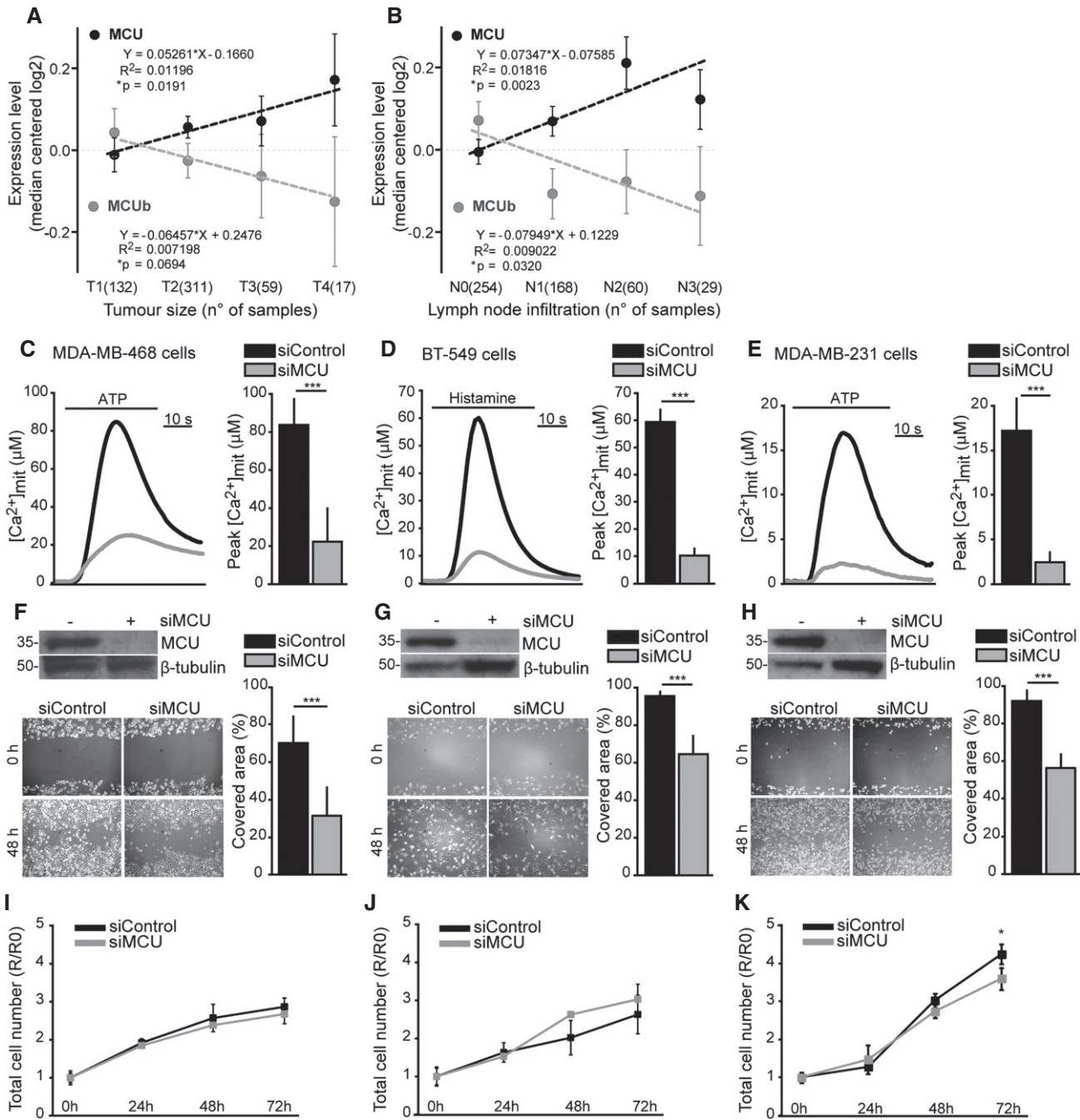

**Figure 1. *MCU* expression correlates with breast tumor progression and TNBC cell migration.**

A, B    Correlation of *MCU* and *MCUb* expression levels with breast cancer clinical stages. Median-centered log2 mRNA expression levels of *MCU* and *MCUb* were collected from the TCGA breast cancer dataset (http://tcga-data.nci.nih.gov/docs/publications/brca_2012/). Data were plotted and analyzed against tumor size (T1–T4) (A) and regional lymph node infiltration (N0–N3) (B), according to the AJCC Cancer Staging Manual (7th edition). Linear regression analysis with different stages was implemented. Parameters of linear regression are shown. Numbers of samples for each stage are shown in parentheses.

C–E    MCU silencing reduces $[Ca^{2+}]_{mit}$ uptake in TNBC cells. Cells were transfected with siMCU or siControl. After 48 h, $[Ca^{2+}]_{mit}$ uptake upon ATP (C, E) or histamine (D) stimulation was measured (n = 10). P-values: ***P = 0.0008 (C), ***P < 0.0001 (D), ***P = 0.0001 (E), respectively.

F–H    MCU silencing impairs TNBC cell migration. Cells were transfected with siMCU or siControl. The day after transfection, a linear scratch was obtained on the cell monolayer through a vertically held P200 tip (time point 0 h). Cell migration into the scratched area was monitored 48 h later. The covered area was measured and expressed as a percentage relative to 0-h time point (n = 12). P-value: ***P < 0.0001.

I–K    Cell proliferation is mainly unaffected by MCU depletion. Cells were transfected with siMCU or siControl. Cell number was counted every 24 h for 3 days (the 72-h time point corresponds to the 48-h time point of wound healing assay). Results are expressed as ratio R/R0 where R0 is the number of cells at the time of transfection (0-h time point) (n = 6). P-value: *P = 0.05.

Data information: In each panel, data are presented as mean ± SD. A two-tailed unpaired *t*-test was performed. See also Appendix Figs S1–S3.

## MCU silencing blunts cell invasiveness without affecting cell viability

To further investigate the molecular mechanism involved in the regulation of migration by MCU, we focused on MDA-MB-231 cells. Of note, re-expression of mouse MCU (Ad-mMCU), in cells in which MCU was silenced, rescued motility confirming the specificity of the effect of siMCU (Fig 2A). Next, the invasion potential of TNBC cells upon MCU silencing was investigated. For this purpose, an *in vitro* spheroid formation assay was performed. Stable MCU-silenced cells were produced and checked for MCU protein downregulation and reduced mitochondrial [$Ca^{2+}$] at rest, and upon agonist stimulation (Appendix Fig S4A–C). shMCU cells were grown in agar containing medium, and spheroid-shaped colonies were moved into a collagen matrix, where they further grew and spread radially into the 3D environment. By monitoring spheroids migration over time, we demonstrated that MCU silencing strongly impairs the ability of TNBC cells to invade the surrounding collagen matrix (Fig 2B). Of note, a colony formation assay revealed that, in 7 days, cell growth was partially inhibited by shMCU (Fig 2C). As already reported (Curry *et al*, 2013), we excluded a role of apoptosis and of cell cycle arrest in our experimental settings (Fig 2D and E). Moreover, the drop in mitochondrial $Ca^{2+}$ uptake upon MCU silencing was not related to alterations in the mitochondrial membrane potential (ΔΨ), as no difference was detected in the steady-state accumulation of the cationic fluorescent dye tetramethyl rhodamine methyl ester (TMRM) in mitochondria (Appendix Fig S4D).

Hence, MCU activity is not limited to the regulation of TNBC cell migration, but it controls the invasion potential of malignant breast cancer cells.

## MCU deletion hampers tumor growth and metastasis formation in MDA-MB-231 xenografts

The *in vitro* data on migration, invasiveness, and clonogenic activity were further supported by an *in vivo* orthotopic tumor analysis. *MCU* deletion of MDA-MB-231 cells was achieved by CRISPR/Cas9 Nuclease RNA-guided genome editing technology (Cong *et al*, 2013). Two independent $MCU^{-/-}$ clones were selected and tested for their reduced resting mitochondrial [$Ca^{2+}$] and agonist-induced $Ca^{2+}$ uptake (Appendix Fig S4E–G), while cytosolic $Ca^{2+}$ transients were unaffected (Appendix Fig S4H). $MCU^{-/-}$ cells were injected into the fat pad of SCID mice, and tumor size, lymph node infiltration, and metastasis formation were measured. Tumor growth was slower in mice injected with $MCU^{-/-}$ cells, relative to controls (Fig 3A). Therefore, mice were sacrificed at different time points to compare the metastatic potential of tumors with equal size (i.e., control mice were sacrificed at day 39 post-injection, while $MCU^{-/-}$ clones 1 and 2 mice were sacrificed at day 46 and 56 p.i., respectively). Independently of tumor size, lymph node infiltration and lung metastasis of $MCU^{-/-}$ tumors were sharply impaired as revealed by *in vivo* imaging of metastasis at the homolateral axillary area (Fig 3B), lymph nodes weight (Fig 3C), lymph nodes infiltration by human cytokeratin-positive cells (Fig 3D), and *ex vivo* imaging of lung metastases (Fig 3E).

These results demonstrate that the molecular knockdown of mitochondrial $Ca^{2+}$ signaling impairs rapid tumor progression and metastasis formation *in vivo*, and well match the data of Fig 1, which showed overexpression of MCU in advanced clinical stages of breast cancer.

## MCU downregulation decreases cellular NADH levels and ATP production, but increases NADPH/NADH ratio

To understand the impact of MCU downregulation on mitochondrial redox metabolism, we measured cellular and mitochondrial NADH levels (the most abundant nicotinamide adenine dinucleotide species present in mitochondria), NADPH levels, and ATP production using live cell fluorescent and luminescent techniques. First, we used a recently developed approach to assess cellular NADPH/NADH homeostasis by discriminating the two autofluorescent species according to their fluorescence lifetime parameters (Blacker *et al*, 2014). By measuring the lifetime component associated with the enzyme-bound fraction of NADPH/NADH ($\tau_{bound}$), the ratio of the two redox equivalents can be directly assessed, while intensity measurements reflect their total amount. Interestingly, in shMCU cells, we observed a significant increase in $\tau_{bound}$, as compared to shControl cells (Fig 4A and B), indicating an increased NADPH/NADH ratio, that was associated with the reduction in total NADPH+NADH intensities in shMCU cells (Fig 4C). Next, to assess the redox ratio of the NADH/$NAD^+$ couple, we compared the resting NADH fluorescence intensity to maximally oxidized (in the presence of the uncoupler carbonyl cyanide4-(trifluoromethoxy)phenylhydrazone, FCCP) and maximally reduced (in the presence of the complex I inhibitor rotenone) state (Fig 4D and E). These measurements indicate that, in spite of the decrease in the total amount of NADH, the redox equilibrium between NADH/$NAD^+$ is unaltered following MCU knockdown. Altogether, these changes suggest a complex alteration in cellular redox state due to the lack of MCU. On the one hand, it implies a mitochondrial bioenergetic defect due to the lack of reducing equivalents used in oxidative phosphorylation (OXPHOS). This has been further demonstrated by measuring ATP production rate after 2-deoxy-D-glucose treatment in MCU-silenced and control cells. Under those settings, MCU silencing significantly reduced mitochondrial ATP production (Fig 4F). On the other hand, the overall increase in NADPH/NADH ratios suggests an augmented cellular antioxidant capacity. This prompted us to investigate further the turnover of mitochondrial reactive oxygen species, which has been previously implicated in regulating cell migration and invasion (Bogeski *et al*, 2011; Sena & Chandel, 2012).

## MCU silencing critically reduces mitochondrial ROS production

It is well established that redox signaling is involved in cellular migration, and a variety of antioxidant molecules have been shown to inhibit cell motility both *in vitro* and *in vivo* (Porporato *et al*, 2014). In line with these observations, treatment of MDA-MB-231 cells with two different antioxidants/reductants (N-acetylcysteine (NAC) and dithioerythritol (DTE)) reduced cell migration, as measured by wound healing assay (Fig 5A). To specifically analyze the effect of mitochondrial ROS on migration, we used the mitochondria-targeted ROS scavenger MitoTEMPO. The effect of Mito-TEMPO on breast cancer cell migration was similar to that obtained by NAC and DTE, thus supporting the hypothesis that mitochondrial ROS play a crucial role in TNBC migration (Fig 5B).

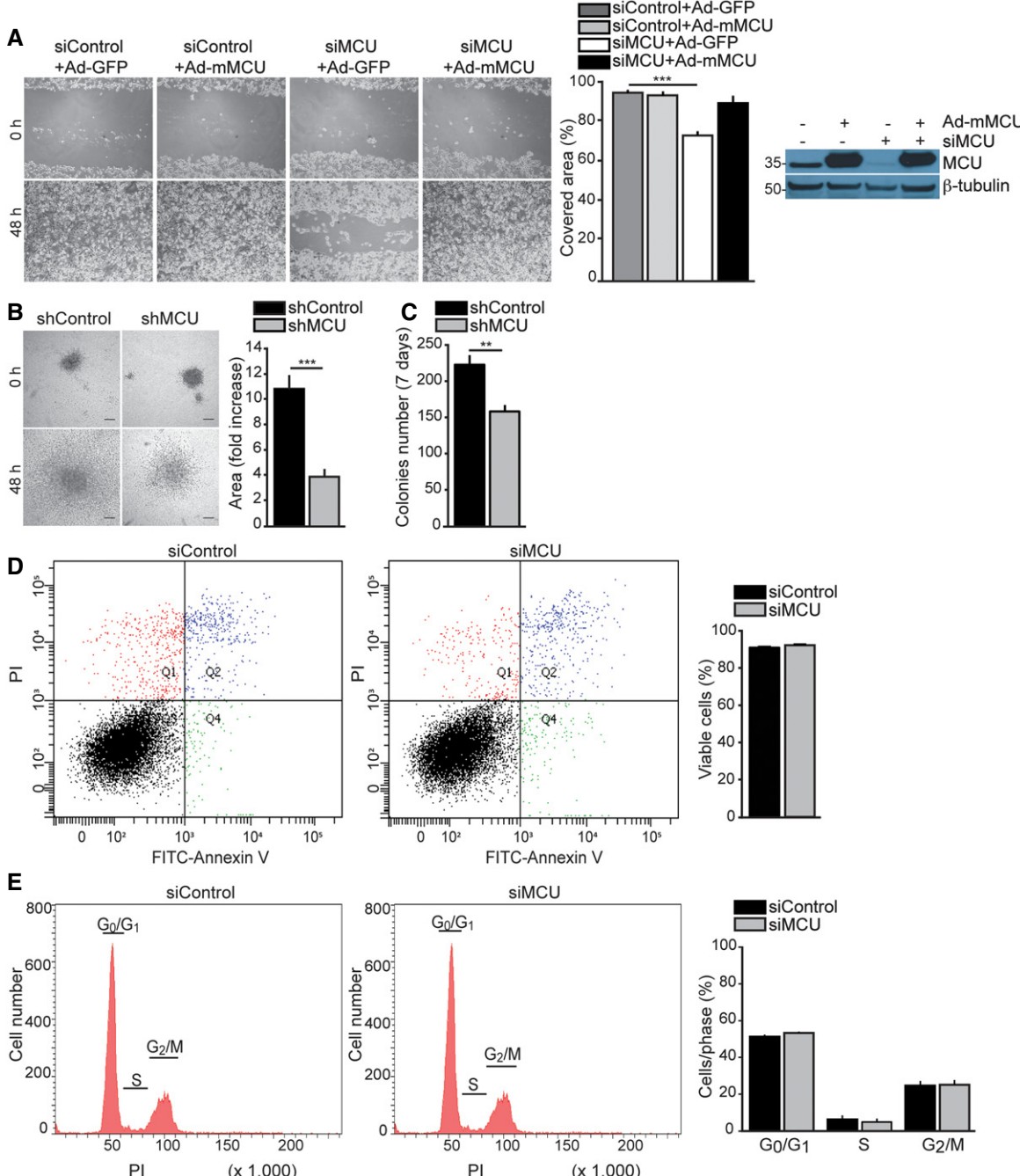

**Figure 2.   MCU silencing blunts cell invasiveness without affecting cell viability.**

A   Re-expression of mMCU rescues cell motility of MCU-silenced cells. Cells were transfected with siMCU or siControl. Ad-mMCU was used to re-express MCU (Ad-GFP was used as a control). MCU protein expression was verified by Western blot. The day after transduction, a linear scratch was made (0-h time point). Cell migration into the wounded area was monitored at 48-h time point, and the covered area was measured ($n = 12$). $P$-value: ***$P < 0.0001$.

B   MCU silencing blunts cell invasiveness. Stable shMCU- and shControl-expressing spheroids were plated and let grow into collagen I (0-h time point). Spheroid area was measured at 0 h and 48 h ($n = 8$). Scale bar: 300 μm. $P$-value: ***$P = 0.0003$.

C   MCU silencing reduces the clonogenic potential of MDA-MB-231 cells. Stable shMCU- and shControl-expressing cells were plated at low confluence ($2 \times 10^3$/well of a 6-well plate). After 7 days, the number of colonies was counted (minimum 30 cells/colony, $n = 8$). $P$-value: **$P = 0.0027$.

D   MCU depletion does not induce cell death. Cells were transfected with siMCU or siControl. Seventy-two hours later, cell apoptosis and necrosis were measured by FITC-Annexin V and propidium iodide (PI) detection (Q1: PI positive, Q2: PI and FITC-Annexin V positive, Q3: PI and FITC-Annexin V negative, Q4: FITC-Annexin V positive; $n = 6$).

E   MCU depletion does not alter cell cycle. Cells were transfected with siMCU or siControl. Seventy-two hours later, cell cycle distribution was monitored by propidium iodide (PI) detection ($n = 6$).

Data information: In each panel, data are presented as mean ± SD. A two-tailed unpaired $t$-test was performed. See also Appendix Fig S4.

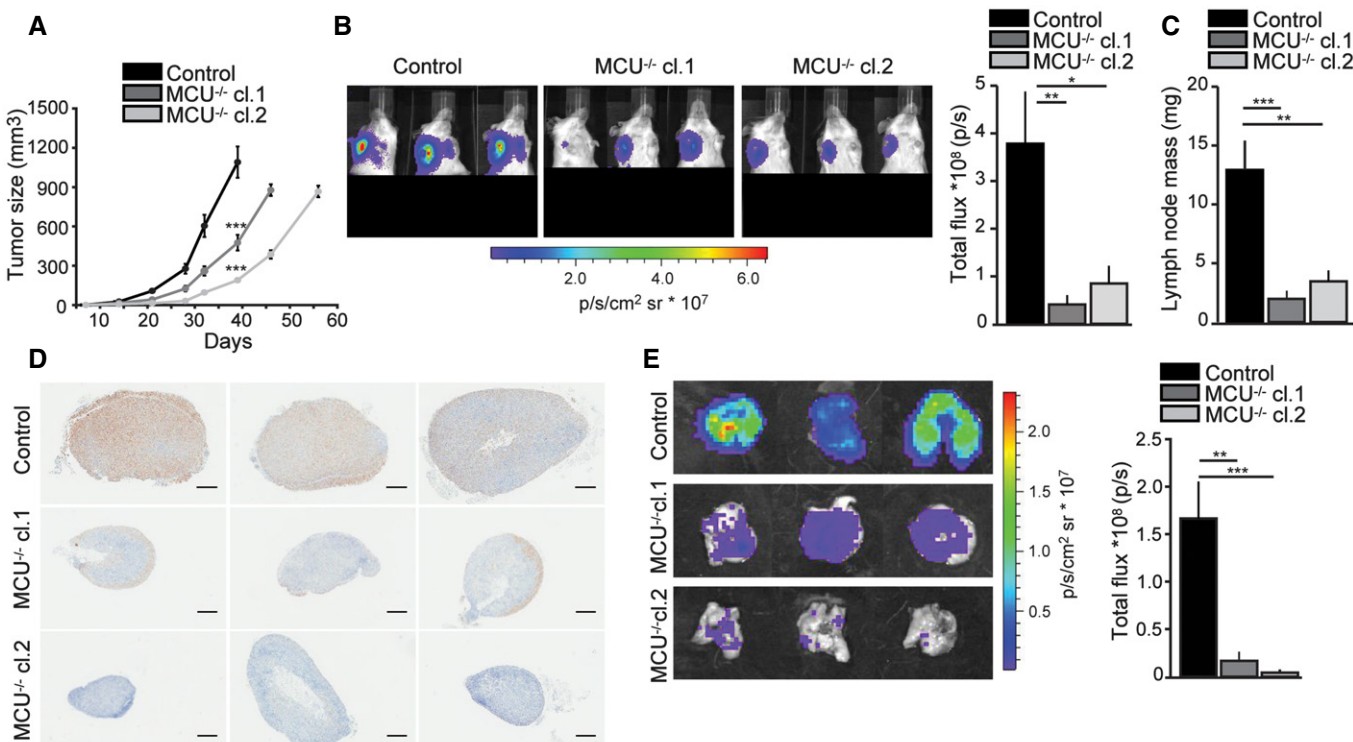

**Figure 3.** *MCU* deletion hampers tumor growth and metastasis formation in MDA-MB-231 xenografts.

Control MDA-MB-231 cells and $MCU^{-/-}$ clones 1 and 2 carrying the firefly luciferase reporter gene were injected into the fat pad of SCID mice.

A   Tumor mass volume was measured at specific time points until the day of sacrifice (day 39 post-injection for control, day 46 and 56 p.i. for $MCU^{-/-}$ cl.1 and cl.2, respectively). *P*-values: (cl.1) ***$P$ = 0.0001, (cl.2) ***$P$ < 0.0001.

B   Left: *in vivo* metastasis at the homolateral axillary area of three representative mice per group at the time of sacrifice. Right: total flux analysis. *P*-values: **$P$ = 0.01, *$P$ = 0.02.

C   Lymph nodes weight at the time of sacrifice. *P*-values: ***$P$ = 0.0010, **$P$ = 0.0014.

D   Human cytokeratin 7 (CK7) IHC staining of three representative lymph nodes per group. Scale bar: 500 μm.

E   Left: images of three representative lungs per group collected *ex vivo* at the time of sacrifice. Right: total flux analysis. *P*-values: **$P$ = 0.0031, ***$P$ = 0.0004.

Data information: In each panel, data are presented as mean ± SE ($n$ = 9 for Control, $n$ = 8 for $MCU^{-/-}$ cl.1, $n$ = 10 for $MCU^{-/-}$ cl.2). A two-tailed unpaired *t*-test was performed. See also Appendix Fig S4.

Next, we sought to verify the role of mitochondrial $Ca^{2+}$ uptake in ROS production. For this purpose, we directly measured mitochondrial hydrogen peroxide ($H_2O_2$) levels with pHyper-dMito protein sensor. One of the major advantages of this probe is that it is ratiometric by excitation, thus limiting measurement errors deriving from photobleaching or concentration variability (Belousov *et al*, 2006). Since pHyper-dMito is known to be sensitive to pH, we in addition measured mitochondrial pH using the redox insensitive form of the sensor, SypHer2 (Shirmanova *et al*, 2015). MCU silencing did not affect matrix pH (Fig 5C) while mitochondrial $H_2O_2$ levels were significantly reduced (Fig 5D). This was further confirmed using two different non-ratiometric redox indicators, the mitochondrial $H_2O_2$-sensitive HyPerRed probe (Ermakova *et al*, 2014) (Fig 5E) and the superoxide anion sensitive dye, MitoSOX™ (Fig 5F). Finally, we took advantage of ectopic expression of mitGrx1-roGFP2, a genetically encoded ratiometric protein sensor for detection of mitochondrial glutathione redox potential ($E_{GSH}$), as a direct indication of oxidative stress (Gutscher *et al*, 2008). Live cell imaging revealed that MCU silencing caused a marked reduction in the GSSG/GSH ratio

(Fig 5G). Altogether, these results show that MCU silencing significantly reduces mitochondrial ROS production, suggesting that mROS may represent the key signaling mediators of MCU-regulated cell motility.

### HIF-1α signaling is a major effector of MCU

One of the main regulators of cell transformation and cancer progression is HIF-1α, which not only plays an essential role in hypoxic tumors, but also regulates a large variety of target genes controlling the malignancy of several tumor types, which express HIF-1α even in normoxic condition (Semenza, 2010). ROS signaling has been reported to increase HIF-1α protein stability (Klimova & Chandel, 2008) and transcription (Movafagh *et al*, 2015), both in normoxic and hypoxic conditions. Given the observed decrease in mROS production by MCU silencing, we asked whether MCU regulates HIF-1α levels, either controlling protein stability or gene transcription. MCU silencing caused a robust downregulation of HIF-1α protein levels (Fig 6A). To understand how siMCU induces HIF-1α depletion, we first investigated the canonical pathway of

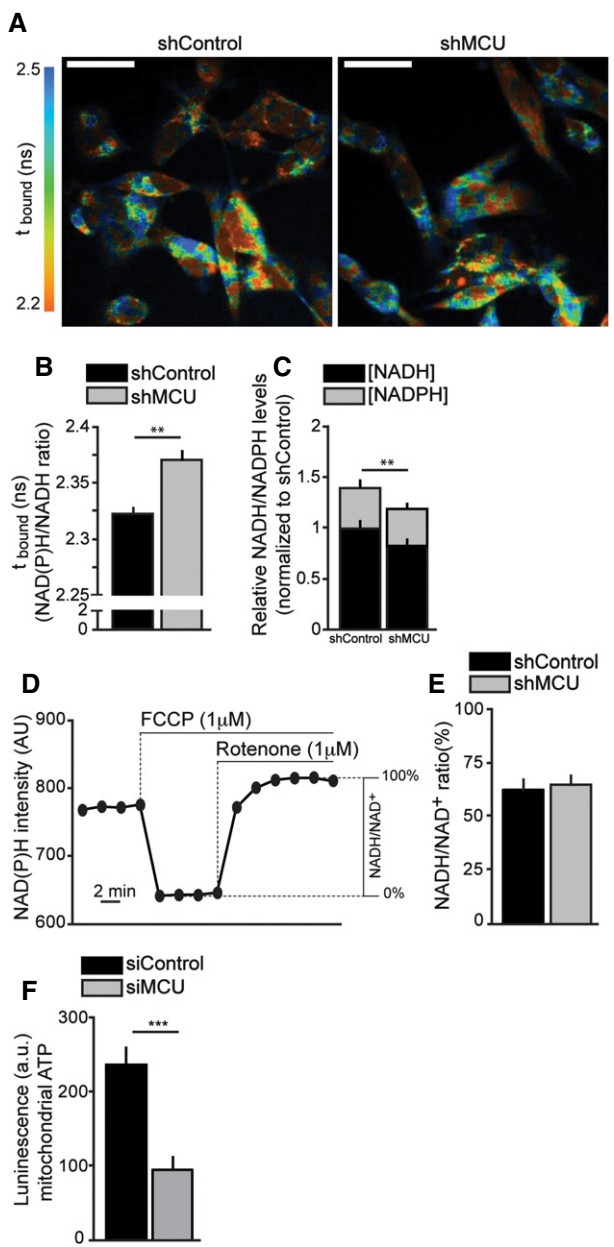

**Figure 4.  MCU downregulation alters cellular redox state.**

A–C  FLIM analysis of cellular NADH/NADPH levels. Fluorescence lifetimes of NAD(P)H autofluorescence in stable shControl- and shMCU-expressing cells were imaged. Representative images of the distribution of $\tau_{bound}$ on an intensity weighted pseudocolored scale (2.2–2.5 ns) are shown. Scale bars: 20 μm (A). Mean ± SE of $\tau_{bound}$ (B) and relative NADH and NADPH intensities (C) calculated from equation in Blacker et al (2014) are shown (n = 3). P-values: **P = 0.01 (B), **P = 0.002 (C).

D, E  Measurement of the redox state of the NADH/NAD$^+$ couple. Representative measurements of NADH intensity at steady state and at minimal and maximal reduced state (D). Percentage of the steady-state redox state (E) (n = 3).

F  MCU depletion impairs the mitochondrial rate of ATP production. Cells were transfected with siMCU or siControl. Forty-eight hours later, cells were treated with 5.5 mM 2-deoxy-D-glucose for 1 h and cellular ATP levels were quantified (n = 6). P-value: ***P = 0.0009.

Data information: In each panel, data are expressed as mean ± SE. A two-tailed unpaired t-test was performed.

HIF-1α protein degradation. Prolyl hydroxylase domain protein 2 (PHD2) hydroxylates HIF-1α in an $O_2$-dependent manner, thus triggering interaction of HIF-1α with von Hippel–Lindau tumor suppressor protein (VHL) and, eventually, proteasome recruitment. We reasoned that, if siMCU enhanced HIF-1α protein degradation, proteasome inhibition would lead to accumulation of hydroxylated HIF-1α (OH-HIF-1α). Thus, we treated MDA-MB-231 cells with the proteasome inhibitor MG132 at different time points and monitored protein levels of both HIF-1α and hydroxylated HIF-1α. As expected, MG132 treatment caused progressive accumulation of HIF-1α and hydroxylated HIF-1α in siControl samples. Surprisingly, both HIF-1α and hydroxylated HIF-1α protein levels were constantly lower after MCU silencing (Fig 6B), suggesting that proteasome-mediated degradation is not responsible for siMCU-dependent HIF-1α depletion.

Thus, we pursued the hypothesis of a transcriptional control of *HIF1A*. Indeed, induction of mitochondrial ROS production by paraquat treatment increased HIF-1α mRNA levels (Fig 6C). In addition, siMCU strongly reduced *HIF1A* transcription both in normoxic and in hypoxic conditions (Fig 6D). Notably, rescue of MCU expression restored HIF-1α mRNA levels (Appendix Fig S5A). Also, *HIF2A* transcription was significantly blunted by siMCU (Fig 6E). Moreover, HIF-1α target genes, selected on the basis of their role in metabolic reprogramming and/or migration control, were induced by hypoxia, as expected (Fig 6F–J). In agreement with HIF-1α downregulation, transcription of these genes was significantly reduced by MCU silencing both in normoxia and in hypoxia (Fig 6F–J). These data indicate that MCU silencing mainly controls transcription of *HIF1A* and of its target genes, presumably through the regulation of mROS production. To verify whether HIF-1α determines shMCU-mediated effects on cell migration, we carried out a rescue experiments by re-expressing HIF-1α in MCU-silenced cells. We observed that HIF-1α overexpression significantly rescues siMCU-mediated impairment of migration (Fig 6K) demonstrating that HIF-1α is a crucial downstream effector of MCU in TNBC. To understand whether similar correlation occurs in human tumors, we analyzed the mRNA levels of HIF-1α and its regulated genes in the TCGA BRCA dataset (see above). Importantly, significant correlations of *MCU* expression with both *HIF1A* and its target genes were found (Fig 6L and M), indicating that MCU-dependent *HIF1A* transcription may also occur in human breast tumors and that MCU may represent a novel regulator of breast cancer progression.

# Discussion

Mitochondrial $Ca^{2+}$ signaling goes far beyond the general stimulation of cellular energetics. In the last decades, the contribution of mitochondrial $Ca^{2+}$ uptake in cell survival and response to apoptotic stimuli has been widely investigated (Rizzuto et al, 2012). The molecular characterization of MCU (Baughman et al, 2011; De Stefani et al, 2011) provided the tools to understand new roles of mitochondrial $Ca^{2+}$ uptake in several pathophysiological conditions, including cancer. By genetic manipulation of MCU complex, the notion that mitochondrial $Ca^{2+}$ signaling is required for cancer progression has emerged (Mallilankaraman et al, 2012; Curry et al, 2013). Indeed, in human breast cancer, a correlation between *MCU* gene expression and poor prognosis has been

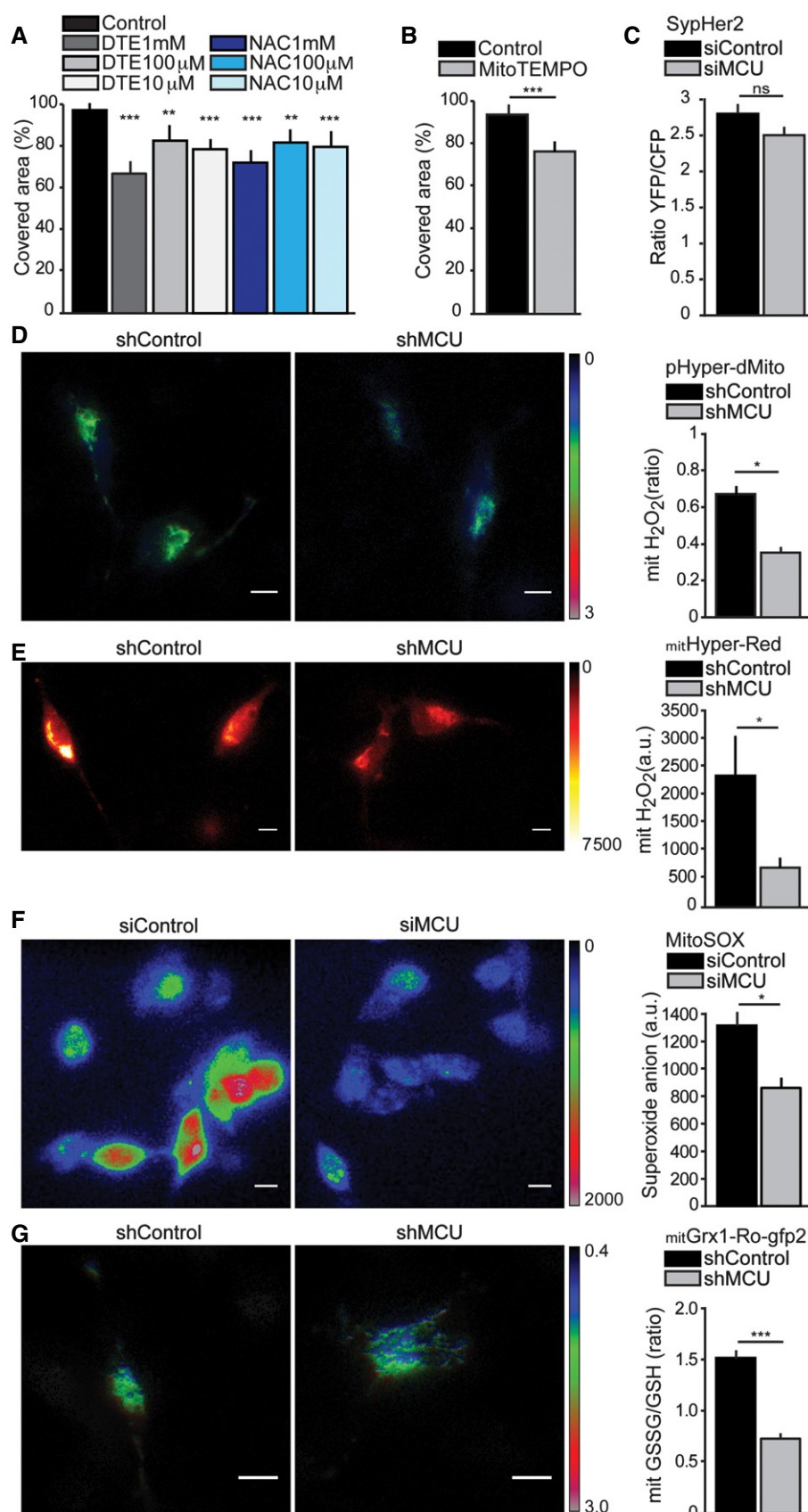

**Figure 5.**

**Figure 5.  MCU depletion reduces mitochondrial ROS production.**

A     Antioxidant treatments decrease cell migration. A linear scratch was obtained on cell monolayer through a vertically held P200 tip (0-h time point). Cells were treated for 48 h with N-acetylcysteine (NAC) or dithioerythritol (DTE). Cell migration into the wounded area was monitored at 48-h time point, and the covered area was measured ($n = 12$). P-values: (DTE 100 μM) **$P = 0.008$, ***$P < 0.0001$, (NAC 100 μM) **$P = 0.005$.

B     Scavenging of mitochondrial ROS decreases cell migration. A linear scratch was obtained on a cell monolayer through a vertically held P200 tip (0-h time point). Cells were treated for 48 h with 50 μM MitoTEMPO. Cell migration into the wounded area was monitored at 48-h time point, and the covered area was measured ($n = 12$). P-value: ***$P < 0.0001$.

C     MCU silencing does not affect matrix pH. Cells were transfected with siMCU or siControl and SypHer2 probe. Forty-eight hours later, matrix pH was measured ($n = 22$).

D, E   Mitochondrial $H_2O_2$ levels are critically blunted after MCU depletion. Cells were transfected with shMCU or shControl, together with the ratiometric YFP-based biosensor pHyper-dMito (D) or the mitochondrial $H_2O_2$-sensitive HyPerRed probe (E). Forty-eight hours later, $H_2O_2$ production was measured ($n = 35$). P-values: *$P = 0.02$ (D), *$P = 0.05$ (E).

F     Mitochondrial superoxide levels are critically blunted after MCU silencing. Cells were transfected with siMCU or siControl. Forty-eight hours later, cells were loaded with the red dye MitoSOX™ and superoxide anion levels were measured ($n = 25$). P-value: *$P = 0.04$.

G     Mitochondrial GSSG/GSH ratio is critically reduced after MCU silencing. Cells were transfected with shMCU or shControl, together with the mitochondrial targeted mitGrx1-roGFP2 probe. Ninety-six hours later, the glutathione redox potential ($E_{GSH}$) was measured ($n = 46$). P-value: ***$P < 0.0001$.

Data information: In panels (A, B), data are expressed as mean ± SD. In panels (C–G), data are expressed as mean ± SE. A two-tailed unpaired $t$-test was performed. Scale bars: 10 μm.

reported (Hall *et al*, 2014). At first sight this evidence may appear in contrast with the previous finding that miR-25, that specifically targets MCU, is expressed in colon and prostate primary tumors (Marchi *et al*, 2013). However, it should be taken into account that metastatic cells must adapt and modify their signaling phenotype and bioenergetic profile, to undergo unrestrained proliferation (LeBleu *et al*, 2014).

On this basis, we investigated the contribution of mitochondrial $Ca^{2+}$ uptake to metastasis. We hypothesized that, while at early stages of tumor formation low mitochondrial $Ca^{2+}$ loading should be preserved to avoid sensitization to apoptotic stimuli (Marchi *et al*, 2013), in advanced stage tumors, high mitochondrial $Ca^{2+}$ levels might have different, favorable roles.

Bioinformatic analysis corroborated this hypothesis indicating a relatively small but strongly significant increase in the expression of the channel forming subunit of the MCU complex during tumor progression. Interestingly, this was accompanied with the reduction in the endogenous dominant-negative *MCUb* isoform. The mRNA levels of the regulatory subunits (*MICU1-3, EMRE*) showed no correlation, suggesting that their posttranslational modification plays more important roles in regulating $Ca^{2+}$ flux through the channel forming MCU subunits (Patron *et al*, 2014; Petrungaro *et al*, 2015).

We thus reasoned that elevated mitochondrial $Ca^{2+}$ transients might be essential for cancer progression. To validate our hypothesis, we chose three different TNBC metastatic cell lines (BT-549, MDA-MB-231, MDA-MB-468) and markedly reduced mitochondrial $Ca^{2+}$ uptake by MCU silencing. We simultaneously monitored the capacity of those cells to migrate and rescue a scratched area. MCU suppression strongly reduced all three TNBC lines migration potential, and this effect could not be justified by short-term changes in cell cycle or death. It has been proposed that MCU regulates store-operated $Ca^{2+}$ entry-dependent cell migration (Tang *et al*, 2015). Our analysis demonstrates a cell line-dependent effect of MCU silencing on cytosolic $[Ca^{2+}]$ and SOCE. One intriguing possibility would be that both increased and decreased cytosolic $[Ca^{2+}]$ similarly regulate cell migration, although with different mechanisms. However, we also show that MCU deletion does not affect cytosolic $[Ca^{2+}]$ in CRISPR/Cas9 clones used for the *in vivo* xenograft experiments. These data convincingly point to a specific role of

mitochondrial $Ca^{2+}$ uptake in the regulation of migration and tumor progression.

Notably, MCU stable depletion reduced cell growth, as demonstrated by the colony formation assay, and the capacity of invading a collagen-based matrix that mimics *in vitro* the potential of metastatic cells to spread into distant tissues. Most importantly, *in vivo* experiments confirmed these results, in terms of primary tumor growth (slower in $MCU^{-/-}$ xenografts), lymph node infiltration, and lung metastasis formation (both parameters being reduced by *MCU* deletion, independently of primary tumor size).

The cellular events that underlie this process are still subject to intensive study and involve a complex rearrangement of mitochondrial and cellular metabolism. In the presence of glucose as nutrient, mitochondrial membrane potential was preserved, while mitochondrial dysfunction (i.e., reduced ATP production) became apparent upon inhibition of glycolysis. In addition, we found a significant reduction in total NAD(P)H following MCU silencing. This cannot be simply explained by a general reduction in the TCA cycle flow, since the redox ratio of the NADH/NAD$^+$ couple remained unchanged. As we already showed in skeletal muscle (Mammucari *et al*, 2015), also in TNBC cells MCU silencing leads to reduced resting mitochondrial $[Ca^{2+}]$, given that the channel is the only source of mitochondrial $Ca^{2+}$ uptake, which is supposed to reduce the activity of three $Ca^{2+}$ sensitive TCA cycle-related enzymes (Rizzuto *et al*, 2012). However, the maintenance of the NADH/NAD$^+$ ratio accompanied by reduced total NAD(P)H levels indicate that (i) either $Ca^{2+}$ has still unknown direct targets in mitochondria (e.g., in NAD(P)H synthetic or transport pathways) or (ii) altered $Ca^{2+}$ homeostasis and TCA activity can be indirectly compensated by altering total cellular redox homeostasis. Notwithstanding the exact mechanism, a crucial consequence of MCU silencing is an increased NAD(P)H/NADH ratio, which has profound consequences on cellular antioxidant capacity. On this basis, we considered that the lack of MCU overall can result in reduced steady-state levels of mitochondrial reactive oxygen species (mROS). ROS are critical triggers of metastasis, both *in vitro* and *in vivo* (Santner *et al*, 2001; Porporato *et al*, 2014) and antioxidant treatments result in migration impairment (Tochhawng *et al*, 2013; Cierlitza *et al*, 2015), as we confirmed in our model. In TNBC cells, mROS production was significantly blunted upon MCU silencing, as demonstrated by

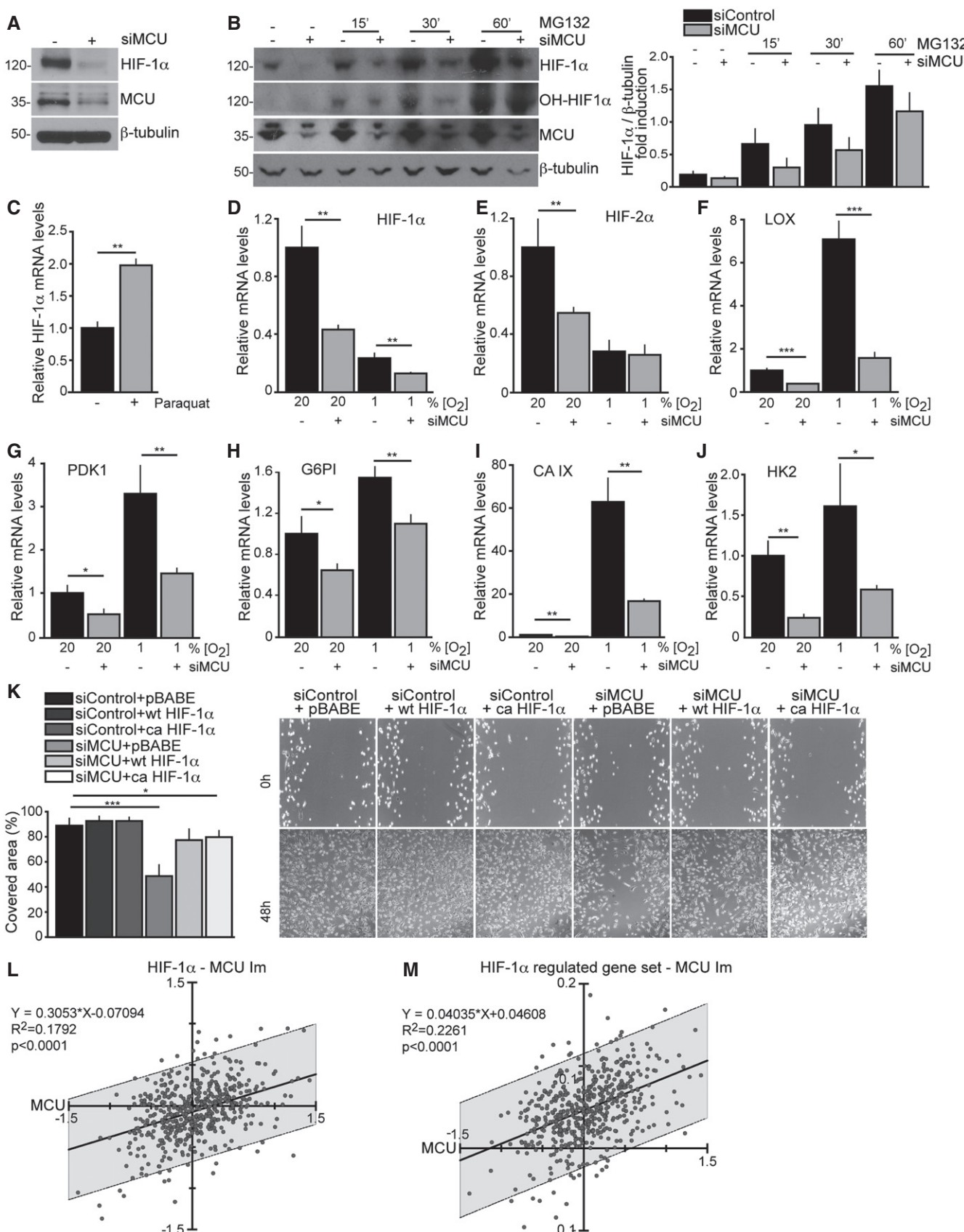

Figure 6.

**Figure 6.  MCU depletion critically affects HIF-1α levels and signaling.**

A    MCU silencing reduces HIF-1α protein levels. Cells were transfected with siMCU or siControl. HIF-1α protein levels were detected 48 h later.

B    MCU silencing reduces MG132-mediated HIF-1α and hydroxylated HIF-1α protein accumulation. Cells were transfected with siMCU or siControl. Forty-eight hours later, cells were treated with 10 μM of the proteasome inhibitor MG132. Left: Protein levels were revealed by Western blot. Right: quantification by densitometry (n = 5).

C    ROS increase *HIF1A* transcription. Cells were treated overnight with 100 μM paraquat to induce ROS production. HIF-1α mRNA levels were measured by real-time PCR (n = 3). *P*-value: **P = 0.002.

D–J    MCU silencing reduces mRNA levels of *HIF1A*, *HIF2A*, and HIF-1α target genes. Cells were transfected with siMCU or siControl. mRNA expression was measured by real-time PCR (n = 3). *P*-values: for HIF-1α **P = 0.0031 (20% O$_2$), **P = 0.009 (1% O$_2$); for HIF-2α **P = 0.01; for LOX ***P = 0.001 (20% O$_2$), ***P = 0.0005 (1% O$_2$); for PDK1 *P = 0.02, **P = 0.009; for G6PI *P = 0.02, **P = 0.005; for CAIX **P = 0.0026 (20% O$_2$), **P = 0.0022 (1% O$_2$); for HK2 **P = 0.0024, *P = 0.03.

K    HIF-1α overexpression rescues siMCU-mediated migration impairment. Cells were transfected with siMCU or siControl. Wild-type (wt) and constitutively active (ca) HIF-1α were expressed by retroviral infection (pBABE was used as a control). The day after transduction, cells were scratched (0-h time point). Cell migration into the wounded area was monitored at 48-h time point, and the covered area was measured (n = 12). *P*-values: ***P < 0.0001, *P = 0.04.

L, M    *MCU* expression levels correlate with *HIF1A* (L) and HIF-1α-regulated genes (M). A linear model (lm) to test the power of *MCU* expression levels predicting the expression of *HIF1A* and HIF-1α-regulated genes was calculated and plotted for each of the 532 samples of the TCGA database (see Fig 1). Equation and $R^2$ values of the linear regression and significance indicating deviation from 0 are shown. The area of 95% prediction limit is shaded below and above the linear regression line. The HIF-1α-regulated gene set was compiled from Broad Institute GSEA database (http://www.broadinstitute.org/gsea/msigdb/cards/V$HIF1_Q5.html, merged sets of V$HIF1_Q3 and V$HIF1_Q5).

Data information: In each panel, data are expressed as mean ± SD. A two-tailed unpaired *t*-test was performed. See also Appendix Fig S5.

accurate analysis of mitochondrial redox state, performed by means of four different mitochondria-targeted redox-sensitive probes. Thus, the final outcome of MCU silencing depends on alterations in the redox potential, which could in turn involve a large number of intracellular signaling cascades. However, our results reveal a smaller effect of mROS depletion on migration, compared to MCU knockdown, indicating that mROS are critical effectors of MCU/Ca$^{2+}$ regulation of metastasis, but most likely cooperate with other yet unresolved mitochondrial signaling molecules.

Recent findings indicate mROS as crucial regulators of protein stabilization and transcription of *HIF1A*, one of the master regulators of tumor progression (Sullivan & Chandel, 2014; Movafagh *et al*, 2015). We show here that, in MDA-MB-231 cells, paraquat treatment, which triggers superoxide production, increases *HIF1A* transcription. This result, together with the evidence that MCU silencing decreases mROS production, prompted us to consider HIF-1α as possible effector of *MCU* depletion. We demonstrated a proteasome-independent regulatory mechanism based on downregulation of *HIF1A* transcription by MCU silencing. Notably, in many solid tumors and cell lines, including MDA-MB-231, *HIF1A* has been reported to be expressed also in normoxic conditions (Hiraga *et al*, 2007). The fact that MCU depletion decreases HIF-1α-dependent transcription also in normoxic conditions suggests that MCU plays a fundamental role to suppress HIF-1α-dependent metabolic reprogramming and migration. HIF-1α is known to induce expression of many different genes that control the wound repair process, as well as metabolic proteins and adhesion proteins (integrins). In cancer cells, HIF-1α induces the expression of several glycolytic protein isoforms that differ from those found in non-malignant cells, including glucose transporters and a plethora of enzymes (Semenza, 2010). In addition to the well-known role of these proteins in promoting the metabolic reprogramming of cancer cells, some HIF-1α-induced glycolytic isoforms also participate in survival processes, including inhibition of apoptosis (i.e., *HKII*) (Sato-Tadano *et al*, 2013) and promotion of cell migration (i.e., *G6PI*) (Torimura *et al*, 2001). Moreover, HIF-1α upregulates lysyl-oxidase (*LOX*) which, in breast cancers, controls migration and invasion (Payne *et al*, 2005). An additional HIF-1α-regulated gene is the carbonic anhydrase *CAIX*, which has been identified as a marker of aggressive carcinomas (Chiche *et al*, 2013). In order to gain insight

into the mechanism, we measured the expression levels of various HIF-1α target genes (namely *PDK1*, *HKII*, *G6PI*, *LOX*, and *CAIX*) and found that MCU silencing counteracts HIF-1α-dependent gene expression.

Finally, HIF-1α overexpression rescues MCU silencing-induced migration impairment, suggesting that HIF-1α represents the key effector of the siMCU-mediated phenotype. Accordingly, we show here that a positive correlation of *MCU* expression with *HIF1A* and its regulated genes exists in human breast cancer samples, indicating that, in parallel with HIF-1α, MCU represents a novel marker of cancer progression.

Overall, our results demonstrate that mitochondrial Ca$^{2+}$ uptake is required for TNBC progression *in vivo*, and clarify the close correlation between mitochondrial Ca$^{2+}$ uptake and mROS production, which targets the transcriptional regulation of *HIF1A*. According to our model, mitochondrial Ca$^{2+}$ uptake prompts sustained mROS production and thus activation of a HIF-1α signaling route that contributes to tumor growth and metastasis formation. This scenario suggests that mitochondrial Ca$^{2+}$ uptake may represent a novel therapeutic target for clinical intervention in aggressive cancers.

## Materials and Methods

### Cell culture and transfection

BT-549 and MDA-MB-468 cells were cultured in Dulbecco's modified Eagle's medium (DMEM) (Life Technologies), supplemented with 10% fetal bovine serum (FBS) (Life Technologies). MDA-MB-231 cells were cultured in DMEM/F12 medium (1:1) (Life Technologies), supplemented with 10% FBS. MCF10AT1k.cl2 and MCF10CA1a.cl1 cells were cultured in DMEM/F12 supplemented with 5% horse serum (HS), 10 μg/ml insulin, 20 ng/ml EGF, 8.5 ng/ml cholera toxin, 500 ng/ml hydrocortisone. All media were supplemented with 1% penicillin G-streptomycin sulfate (Euroclone) and 1% ʟ-glutamine (Euroclone). Cells were maintained in culture at 37°C, with 5% CO$_2$. For experiments performed in hypoxic conditions, cells were cultured for 24 h in a modular incubator chamber at 37°C, with 5% CO$_2$, 94% N$_2$, and 1% O$_2$. O$_2$ levels

were monitored by LabQuest2-Interface and Oxygen Sensor (ML Systems). All cell lines were tested for mycoplasma contamination. siRNAs (10 pmoles/cm$^2$) were transfected using Lipofectamine® RNAiMAX Transfection Reagent (Life Technologies). Expression plasmids were transfected using LT1 reagent (Mirus).

### siRNA

The following MCU-targeting sequences were designed:
siRNA-MCU#1: 5′-GCCAGAGACAGACAAUACUtt-3′.
siRNA-MCU#2: 5′-UAAUUGCCCUCCUUUAUAUtt-3′.

### Expression vectors

The following plasmids were used: pLPCXmitGrx1-roGFP2 and HyperRed, pHyPer-dMito (Evrogen), pLKO.1puro-NonTarget shRNA Control (Sigma-Aldrich), pCMV-VSV-G (a gift from B. Weinberg, Addgene plasmid #8454), pMD2.G (a gift from D. Trono, Addgene plasmid #12259), pLKO.1-TRC cloning vector (a gift from D. Root, Addgene plasmid #10878), HA-HIF1alpha-wt-pBABE-puro and HA-HIF1alpha P402A/P564A-pBABE-puro (gifts from W. Kaelin, Addgene plasmids #19365 and #19005), pBABE-puro (a gift from H. Land & J. Morgenstern & B. Weinberg, Addgene plasmid #1764), and pCL-Eco (a gift from I. Verma, Addgene plasmid #12371).

For MCU stable knockdown in MDA-MB-231, the following interfering sequences were cloned into pLKO.1-TRC cloning vector according to manufacturer's protocol (Addgene):
pLKO.1shMCU#1:
FOR: 5′-CCGGGCAAGGAGTTTCTTTCTCTTTCTCGAGAAAGAGAAA GAAACTCCTTGCTTTTTG-3′
REV: 5′-AATTCAAAAAGCAAGGAGTTTCTTTCTCTTTCTCGAGAAA GAGAAAGAAACTCCTTGC-3′
pLKO.1shMCU#2:
FOR: 5′-CCGGTCAAAGGGCTTAGTGAATATTCTCGAGAATATTCAC TAAGCCCTTTGATTTTTG-3′
REV: 5′-AATTCAAAAATCAAAGGGCTTAGTGAATATTCTCGAGAA TATTCACTAAGCCCTTTGA-3′
For 4mtGCaMP6f cloning, we took advantage of the last generation of GCaMP probes (Chen et al, 2013). cDNA of the probe was amplified from the pGP-CMV-GCaMP6f plasmid, a gift from Douglas Kim (Addgene plasmid # 40755) with the following primers: AAGCTTGGTTCTCATCATCATCATC and GGATCCTCACTTCGCTGT CATCATT and cloned into HindIII and BamHI sites of a custom-made pcDNA3.1-4mt vector.

### Viral infection

Ad-cytAEQ, Ad-mtAEQmut, Ad-GFP, and Ad-MCU were already published (Ainscow & Rutter, 2001; Raffaello et al, 2013).

Lentiviral particles were produced by co-transfection of recombinant shuttle vectors (pCMV8.74 and pMD2.VSVG) and pLKO.1shMCU (#1 and #2) in packaging HEK293T cells. Infected cells were selected by treatment with 1 μg/ml puromycin.

For HIF-1α overexpression in MDA-MB-231 cells, retroviral particles were produced by co-transfection of recombinant shuttle vector pCL-Eco and pBABE vectors (pBABE-puro, HA-HIF-1α-wt-pBABE, HA-HIF-1α-P402A/P564A-pBABE).

### Generation of $MCU^{-/-}$ MDA-MB-231 cell lines

To generate $MCU^{-/-}$ MDA-MB-231 cell lines, two Cas9 guides targeting different regions of the human MCU gene were designed (TGGCGGCTGACGCCCAGCCC for clone1 and GATCGCTTCCTGG CAGAATT for clone2) and cloned into the BsmBI site of the Lenti-CrisprV2 plasmid, a kind gift from Feng Zhang (Addgene plasmid #52961). MDA-MB-231 cells were infected with lentiviral particles produced as described above and selected with puromycin for one week. Dilution cloning was performed to obtain different mono-clonal cell populations that were screened and validated for MCU gene ablation by Western blot. LentiCrisprV2 plasmid was used to produce control clones.

### Antibodies

The following antibodies were used: anti-MCU (1:1,000, HPA016480, Sigma-Aldrich), anti-β-tubulin (1:5,000, sc9104, Santa Cruz), anti-HIF-1α (1:500, 610958, Becton Dickinson), and anti-hydroxy-HIF-1α (1:1,000, 3434, Cell Signaling).

### Ca$^{2+}$ measurements

For measurements of [Ca$^{2+}$]$_{cyt}$ and [Ca$^{2+}$]$_{mit}$, cells grown on 12-mm round glass coverslips were infected with cytosolic (Ad-cytAEQ) or low-affinity mitochondrial (Ad-mtAEQmut) probes, as described (Bonora et al, 2013). Forty-eight hours later, cells were incubated with 5 μM coelenterazine for 2 h in Krebs-Ringer modified buffer (KRB) (125 mM NaCl, 5 mM KCl, 1 mM Na$_3$PO$_4$, 1 mM MgSO$_4$, 5.5 mM glucose, 20 mM HEPES [pH 7.4]) at 37°C supplemented with 1 mM CaCl$_2$, and transferred to the perfusion chamber, and Ca$^{2+}$ transients were evoked by agonist treatments. All aequorin measurements were carried out in KRB, and agonists were added to the same medium.

For SOCC activity measurements, cells grown on 12-mm round glass coverslips were infected with Ad-cytAEQ. Forty-eight hours later cells were incubated with coelenterazine, as described above, and transferred to the perfusion chamber. After 1 min of perfusion with 100 μM EGTA in KRB, agonists and other drugs were added for 2 min, in order to empty intracellular Ca$^{2+}$ stores. Next, cells were perfused with KRB containing 2 mM Ca$^{2+}$ together with agonist or drugs, as indicated.

All aequorin experiments were terminated by lysing the cells with 100 μM digitonin in a hypotonic Ca$^{2+}$-rich solution (10 mM CaCl$_2$ in H$_2$O), thus discharging the remaining aequorin pool. The light signal was collected and calibrated into [Ca$^{2+}$] values as previously described (Bonora et al, 2013).

For measurements of resting mitochondrial [Ca$^{2+}$], cells were grown on 24-mm coverslips and transfected with plasmids encoding 4mtGCaMP6f. After 24 h, coverslips were placed in 1 ml of KRB and imaging was performed on a Zeiss Axiovert 200 microscope equipped with a 40×/1.4 N.A. PlanFluar objective. Excitation was performed with a DeltaRAM V high-speed monochromator (Photon Technology International) equipped with a 75 W xenon arc lamp. Images were captured with a high-sensitivity Evolve 512 Delta EMCCD (Photometrics). The system is controlled by MetaFluor 7.5 (Molecular Devices) and was assembled by Crisel Instruments. In order to perform quantitative

measurements, we took advantage of the isosbestic point in the GCaMP6f excitation spectrum: we experimentally determined in living cells that exciting GCaMP6f at 410 nm leads to fluorescence emission, which is not $Ca^{2+}$ dependent. As a consequence, the ratio between 474-nm and 410-nm excitation wavelengths is proportional to $[Ca^{2+}]$ while independent of probe expression (Hill *et al*, 2014). Cells were thus alternatively illuminated at 474 and 410 nm, and fluorescence was collected through a 515/30-nm band-pass filter (Semrock). Exposure time was set to 200 ms at 474 nm and to 400 ms at 410 nm, in order to account for the low quantum yield at the latter wavelength. At least 15 fields were collected per coverslip, and each field was acquired for 10 s (1 frame/s). Analysis was performed with the Fiji distribution of ImageJ (Schindelin *et al*, 2012). Both images were background corrected frame by frame by subtracting mean pixel values of a cell-free region of interest. Data are presented as the mean of the averaged ratio of all time points.

## Mitochondrial membrane potential (ΔΨ) measurements

Cells were incubated with 20 nM tetramethyl rhodamine methyl ester dye (TMRM) (Life Technologies) for 20 min at 37°C. TMRM fluorescence was measured by FACS. The probe was excited at 560 nm, and the emission light was recorded in the 590–650 nm range; 10 μM FCCP (carbonyl cyanide-*p*-trifluoromethoxyphenyl-hydrazone), an uncoupler of oxidative phosphorylation, was added after 12 acquisitions to completely collapse the ΔΨ. Data are expressed as difference of TMRM fluorescence before and after FCCP depolarization.

## Measurements of NADH/NADPH levels and redox state

For fluorescence lifetime measurements, cells were plated onto 22-mm glass coverslips and allowed to adhere overnight before imaging. At the microscope, coverslips were held at 37°C in a metal ring and bathed in Dulbecco's modified Eagle's medium (Gibco) containing 25 mM glucose, 1 mM pyruvate, and 2 mM glutamine, buffered by 10 mM HEPES; 720-nm two-photon excitation from a Chameleon (Coherent) Ti:sapphire laser was directed through an upright LSM 510 microscope (Carl Zeiss) with a 1.0 NA 40× water-dipping objective. A 650-nm short-pass dichroic and 460 ± 25 nm emission filter separated NAD(P)H fluorescence from the incident illumination. On-sample powers were kept below 10 mW, and emission events were registered by an external detector (HPM-100, Becker & Hickl) attached to a commercial time-correlated single-photon counting electronics module (SPC-830, Becker & Hickl). Scanning was performed continuously for 2 min with a pixel dwell time of 1.6 μs. Subsequent NAD(P)H FLIM data analysis was performed using the procedures detailed in Blacker *et al* (2014).

For measuring NAD(P)H redox state, cells were plated as described above and imaged using a Zeiss 510 META UV-VIS confocal microscope. The blue autofluorescence emitted by the pyridine nucleotides NADH and NADPH in their reduced form was excited with a UV laser (Coherent; at 351 nm), and emission was collected using a 435-nm to 485-nm band-pass filter. To measure the dynamic range of the signal in relation to the fully reduced and oxidized NAD(P)H pool, cells were exposed to carbonyl cyanide

4-(trifluoromethoxy) phenylhydrazone (FCCP [1 μM] to stimulate respiration and achieve maximum NAD(P)H oxidation) and rotenone ([5 μM] to inhibit respiration and achieve maximum NAD(P)H reduction). The final formula used to normalize the NAD(P)H autofluorescence measurements was $(F - F_{FCCP})/(F_{rotenone} - F)$. Quantitative analysis of the images obtained was done using the ImageJ software (http://imagej.nih.gov/ij/).

## ROS production measurements

To determine mitochondrial superoxide levels, cells were loaded with 2 mM MitoSOX™ Red reagent (Life Technologies) for 15 min at 37°C. Images were taken on an inverted microscope (Zeiss Axiovert 200) equipped with a PlanFluar 60×/1.4 NA objective, a Photometrics Evolve Delta EMCCD, and a 75 W Xenon arc lamp coupled to a monochromator (PTI Deltaram V). The system was assembled by Crisel Instruments. MitoSOX™ Red excitation was performed at ~510 nm, and emission was collected at 580 nm. Maximal ROS production was induced with 2.5 μM Antimycin-A (Sigma-Aldrich). Images were taken every 10 s with a fixed 200 ms exposure time. Data were analyzed by ImageJ software.

To determine GSSG/GSH and $H_2O_2$ levels, cells were transfected with plasmids encoding HyperRed, pLPCXmitGrx1-roGFP2, and pHyPer-dMito. To measure mitochondrial pH, SypHer2 plasmid was used. SypHer2 originates from a mutation in a cysteine residue of HyPer that renders it insensitive to $H_2O_2$ but does not affect the pH sensitivity. Images were acquired every 5 s using a Cell Observer High Speed (Zeiss) microscope equipped with 40× oil Fluar (N.A. 1.3) objective, CFP (Semrock HC), YFP and RFP (Zeiss) single-band filters, 420 and 505 nm LED's (Colibri, Zeiss), and an Evolve 512 EMCCD camera (Photometrics). Maximal ROS production was induced with 100 μM $H_2O_2$. To calculate fluorescence ratios, background intensity was subtracted and images were corrected for linear crosstalk. pHyPer-dMito and pLPCXmitGrx1-roGFP2 ratios were calculated by AxioVision software (Zeiss) and analyzed in Excel (Microsoft). HyperRed fluorescence was analyzed by ImageJ software.

## Cell death and cell cycle detection

Cell cycle and cell death induction after MCU silencing were measured by cytofluorometry. Apoptotic and necrotic cells were identified by labeling with FITC-Annexin V (Roche) and propidium iodide (Sigma) for 15 min at 37°C and analyzing cells by FACS (FACS Canto II, BD BioSciences). Data were processed using the BD Vista software.

## Wound healing migration assay

For wound healing assays, cells were seeded at low confluency (30%) in 6-well plates, transfected with siRNA, and cultured in medium without serum. The day after, a linear scratch was obtained on cell monolayers through a vertically held P200 tip and medium was replaced. Images were taken at the indicated time points (time 0 as reference). "TScratch" software (www.cse-lab.ethz.ch/software/) was used for automated image analysis.

    

## Clonogenic assay

To evaluate clonogenic potential, transduced cells were counted and seeded ($10^2$ cells/cm$^2$). Colonies were counted 7 days later. Only, colonies containing $\geq 30$ cells were counted.

## ATP production measurements

ATP production was measured with the ATPlite 1 step Luminescence Assay System (PerkinElmer) according to manufacturer's instructions. Glycolysis was inhibited by treatment with 5.5 mM 2-deoxy-D-glucose for 1 h.

## Spheroids formation assay

$15 \times 10^2$ cells/cm$^2$ were seeded in a 96-well plate containing 100 μl of 1.5% agar in PBS. Seventy-two hours later, spheroids were harvested and collected in tubes filled with 1 ml of medium. Each tube contained five spheroids. Spheroids were let settle to the bottom of the tube, and medium was then sucked out. Spheroids were resuspended in 400 μl/well of a collagen mix solution (1.66 mM L-glutamine, 10% FBS, 0.213% NaHCO$_3$, 1% Pen/Strep, 2 mg/ml Collagen I Bovine Protein (Life Technologies) in MEM (Life Technologies)) and seeded in a 24-well plate, previously filled with 300 μl/well of collagen mix solution. After collagen mix solidification, 1 ml of medium was added in each well. Images were collected every day, for 3 days, and the area of the spheroid cluster was analyzed by Fiji ImageJ software (time 0 as reference).

## RNA extraction, reverse transcription, and quantitative real-time PCR

RNA was extracted using the SV Total RNA Isolation Kit (Promega) following manufacturer's instructions. Complementary DNA was generated with a cDNA synthesis kit (SuperScript II, Invitrogen) and analyzed by real-time PCR (Bio-Rad). HPRT-1 and GAPDH were used as housekeeping genes. The following primers were used:

**HIF-1α:** FOR: TGTACCCTAACTAGCCGAGGAA_ REV: AATCAGC ACCAAGCAGGTCATA

**HIF-2α:** FOR: AATGCAGTACCCAGACGGATTT_ REV: ATGTTTGTC ATGGCACTGAAGC

**LOX:** FOR: TCAGATTTCTTACCCAGCCGAC_ REV: TTGGCATCAAG CAGGTCATAGT

**PDK1:** FOR: AATGCAAAATCACCAGGACAGC_ REV: ATTACCCAG CGTGACATGAACT

**G6PI:** FOR: TTACTCCAAGAACCTGGTGACG_ REV: CTACCAGGA TGGGTGTGTTTGA

**CAIX:** FOR: TGGCTGCTGGTGACATCCTA_ REV: TTGGTTCCCCTT CTGTGCTG

**HK2:** FOR: GTGCCCGCCAGAAGACATTA_ REV: TGCTCAGACCTC GCTCCATT

**HPRT-1:** FOR: TGACACTGGCAAAACAATGCA_ REV: GGTCCTTTT CACCAGCAAGCT

**GAPDH:** FOR: GATTCCACCCATGGCAAATTCC_ REV: CCCCACTTG ATTTTGGAGGGAT

## *In vivo* tumor assays

One control and two independent MDA-MB-231 $MCU^{-/-}$ clones were transduced with a lentiviral vector coding for the Firefly Luciferase reporter gene (Breckpot *et al*, 2003).

For orthotopic tumor assay, $10^6$ cells were resuspended in 100 μl DMEM and injected in the fat pad of six-week-old female SCID mice (Charles River Laboratories). The volume of tumor mass was measured by calipering at specific time points. *In vivo* imaging was performed at the day of sacrifice (day 39 post-injection for control, day 46 p.i. for $MCU^{-/-}$ clone1, and day 56 p.i. for $MCU^{-/-}$ clone2). D-Luciferin (Biosynth AG) (150 mg/kg) was injected i.p. to anesthetized animals. The light emitted from the bioluminescent tumors or metastasis was detected using a cooled charge-coupled device camera mounted on a light-tight specimen box (IVIS Lumina II Imaging System; Caliper Life Sciences). Regions of interest from displayed images were identified around metastatic regions, such as lymph nodes and lungs, and were quantified as total photon counts or photon/s using Living Image® software (Xenogen). In some experiments, the lower portion of each animal was shielded before reimaging in order to minimize the bioluminescence from primary tumor. For *ex vivo* imaging, D-Luciferin (150 mg/kg) was injected i.p. immediately before necropsy. The lungs were excised, placed in a Petri plate, and imaged for 5 min. Animals were randomized before experiments, and no blinding was done. Procedures involving animals and their care were in accordance with the Italian law D. L.vo no 26/2014, and the experimental protocol (Authorization n. 8584/2012-PR) was approved by the Italian Ministry of Health.

## Bioinformatics analysis

To evaluate the correlation of the expression of MCU complex components with tumor progression, median-centered log2 mRNA expression levels of MCUa (NM_138357.2), MCUb (CCDC109b, NM_017918.4), MICU1 (NM_006077.3), MICU2 (NM_152726.2), MICU3 (NM_181723.2), and EMRE (SMDT1, NM_033318.4) were compiled from the TCGA breast cancer dataset (http://tcga-data. nci.nih.gov/docs/publications/brca_2012/) (Koboldt *et al*, 2012). Linear regression analysis of individual expression values with the corresponding tumor size (T1–T4) and lymph node (N0–N3) stages was done in GraphPad (GraphPad Software, Inc.).

To quantify correlation of MCU mRNA levels with HIF-1α and a HIF-1α-regulated gene set, linear models have been constructed in *R*, and prediction values were analyzed against MCU expression levels using linear regression analysis (GraphPad). The HIF-1α-regulated gene set was compiled from the Broad Institute GSEA database (merged sets of V$HIF1_Q3 and V$HIF1_Q5, http://www. broadinstitute.org/gsea/msigdb/cards/V$HIF1_Q5.html).

## Constructing linear models to predict correlations between MCU and HIF-1α and a HIF-1α-regulated gene set

To test whether the expression of members of the MCU complex was predictive of HIF-1α expression, two linear models were created. One to predict the expression of HIF-1α from members of the MCU complex and the other to predict the average expression of HIF-1α-regulated genes. Both linear models were found to be highly

statistically significant (*P*-values 5.67e-12 for predicting HIF-1α and 2.48e-22 for predicting HIF-1α-regulated genes), but these models predict only a relatively small amount of the variation in the data with adjusted $R^2$ values of 0.1099 and 0.1927, respectively. In both models, some members of the MCU complex were found to be more predictive than others. For instance, for predicting the expression of HIF-1α, MCU is the most predictive with a *P*-value of the expression of MCU not being relevant for predicting HIF-1α was 3.83e-06, while for predicting the average of HIF-1α-regulated genes, expression of MICU2 was most significant with a *P*-value of 4.57e-12. Output from *R* and detailed explanation can be found at http://blog.yhathq.com/posts/r-lm-summary .html. The full set of results is summarized in the tables below:

Linear model of MCU predicting HIF-1α expression

|  | Estimate | SE | *t*-value | P (> |t|) |
|---|---|---|---|---|
| MCU | 0.50420 | 0.10794 | 4.671 | 3.83e-06 *** |

Linear model of MCU complex predicting HIF-1α-regulated gene expression

|  | Estimate | SE | *t*-value | P (> |t|) |
|---|---|---|---|---|
| MCU | 0.052493 | 0.009344 | 5.618 | 3.17e-08 *** |

### Statistics

For bioinformatics data, statistical analyses are reported above.

For *in vitro* and *in vivo* experiments, statistical analyses were performed using Student's two-tailed non-paired *t*-tests. *P*-values < 0.05 were considered statistically significant and marked with asterisks (**P* < 0.05; ***P* < 0.01; ****P* < 0.001), as indicated in the figure legends. Data are represented as mean ± SD if not indicated otherwise. Statistical tests applied are indicated in the figure legends.

### Sample size determination

Fisher's exact test has been applied to determine the probability of detecting differences in the following end points:
1) *In vivo* studies: a total number of nine mice for each experimental condition were analyzed in order to detect the expected variation in terms of probability of tumor growth and metastasis formation between treatment conditions, with statistical power of 0.85 and assuming a significance threshold corresponding to *P* < 0.05. *A priori* sample size determinations were performed by the GPower3.1 (www.gpower.hhu.de/) software tool and by a simulation based approach.
2) *In vitro* studies: data available in our laboratory to define the variance of the results were adopted. We have assumed a statistical power of 85% and a significance level of *P* < 0.05 applying the Bonferroni correction whenever required.

**Expanded View** for this article is available online.

### Acknowledgements

The authors are grateful to Markus Hoth for helpful discussions and support, to Ildiko Szabò for critical reading of the manuscript, to Diego De Stefani for cloning of 4mtGCaMP6f, to Maria Patron for cloning of *MCU*-targeting LentiCrisprV2 plasmids and to Denis Vecellio Reane for contributing to the visual abstract.

This research was supported by grants from the European Research Council (ERC mitoCalcium, no. 294777 to R.R.); Italian Telethon Foundation (GPP10005A to R.R.); Italian Ministry of Health (Ricerca Finalizzata to R.R.); Italian Ministry of Education, University, and Research (PRIN to R.R., FIRB to R.R., FIRB Futuro in Ricerca RBFR10EGVP_002 to C.M.); NIH (grant 1P01AG025532-01A1 to R.R.); Cariparo and Cariplo Foundations (to R.R.); and the Italian Association for Cancer Research (AIRC) (to R.R.). G.S. is supported by BBSRC, British Heart Foundation, Wellcome Trust, Italian Association of Cancer Research (AIRC), and the University of Padua (PDA). T.B. is an ERASMUS mobility fellow; T.S.B. and M.R.D. are supported by BBSRC. I.B. acknowledges the support of SFB1027 project C4 and the DFG grant BO3643/3-1, and SFB1027 project A2 to Markus Hoth.

### Author contributions

AT, GS, CM, and RR designed experiments and wrote the manuscript. AT performed most of the experiments. RS performed *in vivo* experiments. RBB performed bioinformatics studies. CK contributed to ROS measurements. TSB and TB performed bioenergetics experiments. AR supervised *in vivo* experiments. IB supervised ROS measurements. GS supervised bioinformatics and bioenergetics studies. MRD co-supervised bioenergetics experiments. CM and RR conceived and directed the project.

### Conflict of interest

The authors declare that they have no conflict of interest.

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

### The paper explained

#### Problem

Strong experimental evidence supports the notion that mitochondrial Ca²⁺ accumulation sensitizes to apoptotic challenges, while reduced mitochondrial Ca²⁺ uptake is considered part of the neoplastic phenotype. However, the observation that highly aggressive cancer cells exhibit robust mitochondrial Ca²⁺ responses apparently contradicts this paradigm.

#### Results

Here, we analyzed mitochondrial Ca²⁺ homeostasis in breast cancer. Our results revealed that the expression of MCU, the highly selective channel responsible for mitochondrial Ca²⁺ uptake, correlates with tumor progression. In addition, MCU deletion impairs tumor growth and metastasis formation and inhibits ROS production and HIF-1α expression, two major triggers of cancer progression.

#### Impact

These results disclose a novel role for mitochondrial Ca²⁺ uptake and indicate MCU as a novel druggable target for breast cancer therapy.

Belousov VV, Fradkov AF, Lukyanov KA, Staroverov DB, Shakhbazov KS, Terskikh AV, Lukyanov S (2006) Genetically encoded fluorescent indicator for intracellular hydrogen peroxide. *Nat Methods* 3: 281–286

Blacker TS, Mann ZF, Gale JE, Ziegler M, Bain AJ, Szabadkai G, Duchen MR (2014) Separating NADH and NADPH fluorescence in live cells and tissues using FLIM. *Nat Commun* 5: 3936

Bogeski I, Kappl R, Kummerow C, Gulaboski R, Hoth M, Niemeyer BA (2011) Redox regulation of calcium ion channels: chemical and physiological aspects. *Cell Calcium* 50: 407–423

Bonora M, Giorgi C, Bononi A, Marchi S, Patergnani S, Rimessi A, Rizzuto R, Pinton P (2013) Subcellular calcium measurements in mammalian cells using jellyfish photoprotein aequorin-based probes. *Nat Protoc* 8: 2105–2118

Breckpot K, Dullaers M, Bonehill A, van Meirvenne S, Heirman C, de Greef C, van der Bruggen P, Thielemans K (2003) Lentivirally transduced dendritic cells as a tool for cancer immunotherapy. *J Gene Med* 5: 654–667

Chen TW, Wardill TJ, Sun Y, Pulver SR, Renninger SL, Baohan A, Schreiter ER, Kerr RA, Orger MB, Jayaraman V *et al* (2013) Ultrasensitive fluorescent proteins for imaging neuronal activity. *Nature* 499: 295–300

Chiche J, Ricci JE, Pouyssegur J (2013) Tumor hypoxia and metabolism – towards novel anticancer approaches. *Ann Endocrinol (Paris)* 74: 111–114

Cierlitza M, Chauvistre H, Bogeski I, Zhang X, Hauschild A, Herlyn M, Schadendorf D, Vogt T, Roesch A (2015) Mitochondrial oxidative stress as a novel therapeutic target to overcome intrinsic drug resistance in melanoma cell subpopulations. *Exp Dermatol* 24: 155–157

Cong L, Ran FA, Cox D, Lin S, Barretto R, Habib N, Hsu PD, Wu X, Jiang W, Marraffini LA *et al* (2013) Multiplex genome engineering using CRISPR/Cas systems. *Science* 339: 819–823

Curry MC, Peters AA, Kenny PA, Roberts-Thomson SJ, Monteith GR (2013) Mitochondrial calcium uniporter silencing potentiates caspase-independent cell death in MDA-MB-231 breast cancer cells. *Biochem Biophys Res Commun* 434: 695–700

De Stefani D, Raffaello A, Teardo E, Szabo I, Rizzuto R (2011) A forty-kilodalton protein of the inner membrane is the mitochondrial calcium uniporter. *Nature* 476: 336–340

Elias AD (2010) Triple-negative breast cancer: a short review. *Am J Clin Oncol* 33: 637–645

Ermakova YG, Bilan DS, Matlashov ME, Mishina NM, Markvicheva KN, Subach OM, Subach FV, Bogeski I, Hoth M, Enikolopov G *et al* (2014) Red fluorescent genetically encoded indicator for intracellular hydrogen peroxide. *Nat Commun* 5: 5222

Foskett JK, Philipson B (2015) The mitochondrial Ca uniporter complex. *J Mol Cell Cardiol* 78: 3–8

Gottlieb E, Tomlinson IP (2005) Mitochondrial tumour suppressors: a genetic and molecular update. *Nat Rev Cancer* 5: 857–866

Gutscher M, Pauleau AL, Marty L, Brach T, Wabnitz GH, Samstag Y, Meyer AJ, Dick TP (2008) Real-time imaging of the intracellular glutathione redox potential. *Nat Methods* 5: 553–559

Hall DD, Wu Y, Domann FE, Spitz DR, Anderson ME (2014) Mitochondrial calcium uniporter activity is dispensable for MDA-MB-231 breast carcinoma cell survival. *PLoS ONE* 9: e96866

Hanahan D, Weinberg RA (2011) Hallmarks of cancer: the next generation. *Cell* 144: 646–674

Hill JM, De Stefani D, Jones AW, Ruiz A, Rizzuto R, Szabadkai G (2014) Measuring baseline Ca(2+) levels in subcellular compartments using genetically engineered fluorescent indicators. *Methods Enzymol* 543: 47–72

Hiraga T, Kizaka-Kondoh S, Hirota K, Hiraoka M, Yoneda T (2007) Hypoxia and hypoxia-inducible factor-1 expression enhance osteolytic bone metastases of breast cancer. *Cancer Res* 67: 4157–4163

Klimova T, Chandel NS (2008) Mitochondrial complex III regulates hypoxic activation of HIF. *Cell Death Differ* 15: 660–666

Koboldt DC, Fulton RS, McLellan MD, Schmidt H, Kalicki-Veizer J, McMichael JF, Fulton LL, Dooling DJ, Ding L, Mardis ER *et al* (2012) Comprehensive molecular portraits of human breast tumours. *Nature* 490: 61–70

LeBleu VS, O'Connell JT, Gonzalez Herrera KN, Wikman H, Pantel K, Haigis MC, de Carvalho FM, Damascena A, Domingos Chinen LT, Rocha RM *et al* (2014) PGC-1alpha mediates mitochondrial biogenesis and oxidative phosphorylation in cancer cells to promote metastasis. *Nat Cell Biol* 16: 992–1003, 1001–1015

Mallilankaraman K, Doonan P, Cardenas C, Chandramoorthy HC, Muller M, Miller R, Hoffman NE, Gandhirajan RK, Molgo J, Birnbaum MJ *et al* (2012) MICU1 is an essential gatekeeper for MCU-mediated mitochondrial Ca(2+) uptake that regulates cell survival. *Cell* 151: 630–644

Mammucari C, Gherardi G, Zamparo I, Raffaello A, Boncompagni S, Chemello F, Cagnin S, Braga A, Zanin S, Pallafacchina G *et al* (2015) The mitochondrial calcium uniporter controls skeletal muscle trophism in vivo. *Cell Rep* 10: 1269–1279

Marchi S, Lupini L, Patergnani S, Rimessi A, Missiroli S, Bonora M, Bononi A, Corra F, Giorgi C, De Marchi E *et al* (2013) Downregulation of the mitochondrial calcium uniporter by cancer-related miR-25. *Curr Biol* 23: 58–63

Movafagh S, Crook S, Vo K (2015) Regulation of hypoxia-inducible factor-1a by reactive oxygen species: new developments in an old debate. *J Cell Biochem* 116: 696–703

Owens KM, Kulawiec M, Desouki MM, Vanniarajan A, Singh KK (2011) Impaired OXPHOS complex III in breast cancer. *PLoS ONE* 6: e23846

Patron M, Checchetto V, Raffaello A, Teardo E, Vecellio Reane D, Mantoan M, Granatiero V, Szabo I, De Stefani D, Rizzuto R (2014) MICU1 and MICU2 finely tune the mitochondrial Ca2+ uniporter by exerting opposite effects on MCU activity. *Mol Cell* 53: 726–737

Payne SL, Fogelgren B, Hess AR, Seftor EA, Wiley EL, Fong SF, Csiszar K, Hendrix MJ, Kirschmann DA (2005) Lysyl oxidase regulates breast cancer cell migration and adhesion through a hydrogen peroxide-mediated mechanism. *Cancer Res* 65: 11429–11436

Perocchi F, Gohil VM, Girgis HS, Bao XR, McCombs JE, Palmer AE, Mootha VK (2010) MICU1 encodes a mitochondrial EF hand protein required for Ca(2+) uptake. *Nature* 467: 291–296

Petrungaro C, Zimmermann KM, Kuttner V, Fischer M, Dengjel J, Bogeski I, Riemer J (2015) The Ca(2+)-Dependent Release of the Mia40-Induced MICU1-MICU2 Dimer from MCU Regulates Mitochondrial Ca(2+) Uptake. *Cell Metab* 22: 721–733

Plovanich M, Bogorad RL, Sancak Y, Kamer KJ, Strittmatter L, Li AA, Girgis HS, Kuchimanchi S, De Groot J, Speciner L *et al* (2013) MICU2, a paralog of MICU1, resides within the mitochondrial uniporter complex to regulate calcium handling. *PLoS ONE* 8: e55785

Porporato PE, Payen VL, Perez-Escuredo J, De Saedeleer CJ, Danhier P, Copetti T, Dhup S, Tardy M, Vazeille T, Bouzin C *et al* (2014) A mitochondrial switch promotes tumor metastasis. *Cell Rep* 8: 754–766

Raffaello A, De Stefani D, Sabbadin D, Teardo E, Merli G, Picard A, Checchetto V, Moro S, Szabo I, Rizzuto R (2013) The mitochondrial calcium uniporter is a multimer that can include a dominant-negative pore-forming subunit. *EMBO J* 32: 2362–2376

Rizzuto R, De Stefani D, Raffaello A, Mammucari C (2012) Mitochondria as sensors and regulators of calcium signalling. *Nat Rev Mol Cell Biol* 13: 566–578

Roesch A, Vultur A, Bogeski I, Wang H, Zimmermann KM, Speicher D, Korbel C, Laschke MW, Gimotty PA, Philipp SE *et al* (2013) Overcoming intrinsic multidrug resistance in melanoma by blocking the mitochondrial respiratory chain of slow-cycling JARID1B(high) cells. *Cancer Cell* 23: 811−825

Sancak Y, Markhard AL, Kitami T, Kovacs-Bogdan E, Kamer KJ, Udeshi ND, Carr SA, Chaudhuri D, Clapham DE, Li AA *et al* (2013) EMRE is an essential component of the mitochondrial calcium uniporter complex. *Science* 342: 1379−1382

Santner SJ, Dawson PJ, Tait L, Soule HD, Eliason J, Mohamed AN, Wolman SR, Heppner GH, Miller FR (2001) Malignant MCF10CA1 cell lines derived from premalignant human breast epithelial MCF10AT cells. *Breast Cancer Res Treat* 65: 101−110

Sato-Tadano A, Suzuki T, Amari M, Takagi K, Miki Y, Tamaki K, Watanabe M, Ishida T, Sasano H, Ohuchi N (2013) Hexokinase II in breast carcinoma: a potent prognostic factor associated with hypoxia-inducible factor-1alpha and Ki-67. *Cancer Sci* 104: 1380−1388

Schindelin J, Arganda-Carreras I, Frise E, Kaynig V, Longair M, Pietzsch T, Preibisch S, Rueden C, Saalfeld S, Schmid B *et al* (2012) Fiji: an open-source platform for biological-image analysis. *Nat Methods* 9: 676−682

Sciacovelli M, Gaude E, Hilvo M, Frezza C (2014) The metabolic alterations of cancer cells. *Methods Enzymol* 542: 1−23

Semenza GL (2010) Defining the role of hypoxia-inducible factor 1 in cancer biology and therapeutics. *Oncogene* 29: 625−634

Sena LA, Chandel NS (2012) Physiological roles of mitochondrial reactive oxygen species. *Mol Cell* 48: 158−167

Shirmanova MV, Druzhkova IN, Lukina MM, Matlashov ME, Belousov VV, Snopova LB, Prodanetz NN, Dudenkova VV, Lukyanov SA, Zagaynova EV (2015) Intracellular pH imaging in cancer cells in vitro and tumors in vivo using the new genetically encoded sensor SypHer2. *Biochim Biophys Acta* 1850: 1905−1911

Sullivan LB, Chandel NS (2014) Mitochondrial reactive oxygen species and cancer. *Cancer Metab* 2: 17

Tang S, Wang X, Shen Q, Yang X, Yu C, Cai C, Cai G, Meng X, Zou F (2015) Mitochondrial Ca uniporter is critical for store-operated Ca entry-dependent breast cancer cell migration. *Biochem Biophys Res Commun* 458: 186−193

Tochhawng L, Deng S, Pervaiz S, Yap CT (2013) Redox regulation of cancer cell migration and invasion. *Mitochondrion* 13: 246−253

Torimura T, Ueno T, Kin M, Harada R, Nakamura T, Kawaguchi T, Harada M, Kumashiro R, Watanabe H, Avraham R *et al* (2001) Autocrine motility factor enhances hepatoma cell invasion across the basement membrane through activation of beta1 integrins. *Hepatology* 34: 62−71

Wu WS (2006) The signaling mechanism of ROS in tumor progression. *Cancer Metastasis Rev* 25: 695−705

