## [Review Process File · EMBO Molecular Medicine]

The Mitochondrial Calcium Uniporter regulates breast cancer progression via HIF-1

Anna Tosatto, Roberta Sommaggio, Carsten Kummerow, Robert R. Bentham, Thomas S. Blacker, Tunde Berecz, Michael R. Duchon, Antonio Rosato, Ivan Bogeski, Gyorgy Szabadkai, Rosario Rizzuto and Cristina Mammucari

Corresponding authors: Rosario Rizzuto and Cristina Mammucari, University of Padova

Review timeline:

Submission date:	01 June 2015
Editorial Decision:	23 June 2015
Resubmission:	27 January 2016
Editorial Decision:	15 February 2016
Revision received:	29 February 2016
Accepted:	02 March 2016

Transaction Report:

Editor: Céline Carret

1st Editorial Decision

23 June 2015

Thank you for the submission of your research manuscript to our editorial office. We have now received 3-set of reports, copied below.

As you will see, all three referees agree that technically the paper is well executed, however while also agreeing on its potential interest, referees had a different take on it.

Referees 1 and 2 are more positive, still questioning the novelty aspect of the study, the maybe inappropriate use of the MCU inhibitor and the limited provision of mechanistic understanding. Referee 3 (who knows our journal well) is much more critical and raise serious conceptual issues. For example, the cell line used is not appropriate for looking into metastasis and there seem to be some misconception between migration/invasion and metastasis, especially when the drug used already reduced tumour size. This referee also highlights the study limitations in term of novelty, mechanism and most importantly for our scope, confirms our suspicions that the study lacks medical/clinical validation and pathophysiological insights which we believe, would need more than 3-months to address.

Given the nature of these criticisms and the amount of work likely to be required to address them, I am afraid that we do not feel it would be productive to call for a revised version of your manuscript at this stage and therefore we cannot offer to publish it.

We would, however, have no objection to consider a new manuscript on the same topic if at some time in the near future you obtained data that would considerably strengthen the message of the

study (especially in vivo) and address the referees concerns in full. To be completely clear, however, I would like to stress that if you were to send a new manuscript this would be treated as a new submission rather than a revision and would be reviewed afresh, in particular with respect to the literature and the novelty of your findings at the time of resubmission. If you decide to follow this route, please make sure you nevertheless upload a letter of response to the referees' comments.

At this stage though, I am really sorry to have to disappoint you. I nevertheless hope, that the referee comments will be helpful in your continued work in this area.

***** Reviewer's comments *****

Referee #1 (Comments on Novelty/Model System):

The report that MCU levels can impact on the migratory capacity of cancer cells by controlling ROS levels and HIF1a signalling is new and important. Most of the results were obtained in vitro using appropriate cellular models and assays.

Referee #1 (Remarks):

This study reports that inhibition of the mitochondrial calcium uniporter (MCU) decreases the migration of triple negative breast cancer cells without affecting their proliferation or cell cycle progression. MCU gene silencing or pharmacological inhibition decreased cell migration in the wound healing assay and in collagen matrix and reduced tumour mass volume in mouse xenografts. MCU silencing decreased mitochondrial ATP and ROS production and decreased the transcription of HIF1a and of HIF1a-responsive genes, while enforced expression of HIF1a restored migration of MCU-silenced cells. The authors conclude that MCU-mediated mitochondrial Ca²⁺ uptake boosts the metastatic potential of breast cancer cells by driving the production of mitochondrial ROS that promote HIF1a transcription.

Comments:

The report that MCU levels can impact on the migratory capacity of cancer cells by controlling ROS levels and HIF1a signalling is new and important. The data are clear, the experiments well designed and the results largely consistent with the proposed Ca²⁺/ROS/HIF1a axis. To avoid ambiguities, potential alterations in cytosolic calcium responses should be documented given a previous publication implicating store-operated calcium entry in MCU-dependent cell migration, and the effect of the promiscuous compound used to inhibit the MCU on cytosolic calcium responses should be tested. I also have a few queries for control experiments and clarifications.

1. The effects of MCU silencing on cytosolic Ca²⁺ levels are ignored. MCU silencing was previously shown to decrease breast cancer cell migration by decreasing store-operated Ca²⁺ entry (SOCE) (Tang et al., 2015) an effect that had been proposed early on to reflect the Ca²⁺ buffering ability of mitochondria to relieve the Ca²⁺-dependent inactivation of SOCE Ca²⁺ entry channels. Alteration in cytosolic Ca²⁺ signals, rather than alteration in mitochondrial matrix Ca²⁺, might therefore contribute to the observed phenotype. To clarify this point, recordings of the cytosolic Ca²⁺ levels during stimulation with ATP, as well as quantification of the activity of SOCE channels in control, MCU silenced, and MCU overexpressing cells are required. In order to compare the mitochondrial Ca²⁺ response, the amount of Ca²⁺ stored in the ER should also be assessed by measuring the cytosolic Ca²⁺ response evoked by ATP in a Ca²⁺-free medium, as well as the total amount of stored Ca²⁺ that can be mobilized by ionomycin.

3. The use of KB-R7943 as a "specific MCU inhibitor" (p. 5, last para) is questionable. This isothiourea derivative is the prototypical inhibitor of the reverse mode of the plasma membrane Na⁺/Ca²⁺ exchanger (NCX) (Iwamoto et al., 2007; Iwamoto et al., 1996) and has been used extensively as such to explore the effects of reverse NCX inhibition. Besides this primary action, KB-R7943 was reported to inhibit TRPC channels (Kraft, 2007), L-type voltage-gated Ca²⁺ channels (Ouardouz et al., 2005), ryanodine receptors (Barrientos et al., 2009), NMDA receptors (Brustovetsky et al., 2011), and SOCE channels (Arakawa et al., 2000), while its action as an inhibitor of mitochondrial Ca²⁺ uptake has been questioned (Wiczler et al., 2014). Regardless of

whether it inhibits the MCU or not, KB-R7943 is clearly a promiscuous pharmacologic agent with multiple Ca²⁺-regulatory targets, and considering the long incubation time used here one cannot rely on such a tool as proof of implication of the MCU. If the authors intend to keep these pharmacological data they should at least validate the specificity of the inhibitor for mitochondrial Ca²⁺ uptake vs. other Ca²⁺ transporters by measuring its effects on cytosolic Ca²⁺ responses.

3. Controls are lacking for the measurements of mitochondrial ROS production with Hyper and Hyper red (Fig 4). These probes are pH sensitive and parallel measurements of the matrix pH with dyes or fluorescent proteins are required to rule out that the responses do not reflect a matrix alkalinisation.

4. To establish the causal link between MCU and Hif1a transcription, the authors should show that Hif1a transcription is restored by re-expressing the MCU protein in MCU-silenced cells.

Other points

The quantification of the western blots shown in Fig. 5B are lacking error bars. Was this experiment performed only once?

The original observation that initiated the study is the huge difference in mitochondrial Ca²⁺ uptake between premalignant and malignant cell lines (Fig 1A), but the genetic basis of this difference is not documented. What is the expression level of MCU in premalignant cell line? Is the difference in uptake due to altered MCU levels or to altered level of MCU regulatory proteins?

Related to the point above, the mechanism proposed would be strengthened by showing that the premalignant cell line with low mitochondrial Ca²⁺ uptake migrates poorly and produces low level of mitochondrial ROS.

References cited

- Arakawa, N., Sakaue, M., Yokoyama, I., Hashimoto, H., Koyama, Y., Baba, A. and Matsuda, T. (2000). KB-R7943 inhibits store-operated Ca(2+) entry in cultured neurons and astrocytes. *Biochem Biophys Res Commun* 279, 354-7.
- Barrientos, G., Bose, D. D., Feng, W., Padilla, I. and Pessah, I. N. (2009). The Na⁺/Ca²⁺ exchange inhibitor 2-(2-(4-(4-nitrobenzyloxy)phenyl)ethyl)isothiourea methanesulfonate (KB-R7943) also blocks ryanodine receptors type 1 (RyR1) and type 2 (RyR2) channels. *Mol Pharmacol* 76, 560-8.
- Brustovetsky, T., Brittain, M. K., Sheets, P. L., Cummins, T. R., Pinelis, V. and Brustovetsky, N. (2011). KB-R7943, an inhibitor of the reverse Na⁺ /Ca²⁺ exchanger, blocks N-methyl-D-aspartate receptor and inhibits mitochondrial complex I. *Br J Pharmacol* 162, 255-70.
- Iwamoto, T., Watanabe, Y., Kita, S. and Blaustein, M. P. (2007). Na⁺/Ca²⁺ exchange inhibitors: a new class of calcium regulators. *Cardiovasc Hematol Disord Drug Targets* 7, 188-98.
- Iwamoto, T., Watano, T. and Shigekawa, M. (1996). A novel isothiourea derivative selectively inhibits the reverse mode of Na⁺/Ca²⁺ exchange in cells expressing NCX1. *J Biol Chem* 271, 22391-7.
- Kraft, R. (2007). The Na⁺/Ca²⁺ exchange inhibitor KB-R7943 potently blocks TRPC channels. *Biochem Biophys Res Commun* 361, 230-6.
- Ouardouz, M., Zamponi, G. W., Barr, W., Kiedrowski, L. and Stys, P. K. (2005). Protection of ischemic rat spinal cord white matter: Dual action of KB-R7943 on Na⁺/Ca²⁺ exchange and L-type Ca²⁺ channels. *Neuropharmacology* 48, 566-75.
- Tang, S., Wang, X., Shen, Q., Yang, X., Yu, C., Cai, C., Cai, G., Meng, X. and Zou, F. (2015). Mitochondrial Ca(2+)(+) uniporter is critical for store-operated Ca(2+)(+) entry-dependent breast cancer cell migration. *Biochem Biophys Res Commun* 458, 186-93.

Wiczler, B. M., Marcu, R. and Hawkins, B. J. (2014). KB-R7943, a plasma membrane Na(+)/Ca(2+) exchanger inhibitor, blocks opening of the mitochondrial permeability transition pore. *Biochem Biophys Res Commun* 444, 44-9.

Referee #2 (Remarks):

Manuscript Number: EMM-2015-05491

Title: The Mitochondrial Calcium Uniporter regulates triple negative breast cancer cell migration via HIF-1 suppression

Corresponding Author: Dr. Mammucari

General

This study provides evidence for a functional role of the MCU mitochondrial calcium uniporter, in progression to metastasis in TNBC triple negative breast cancer cell models. The study shows a relation between MCU expression, mitochondrial ROS species and HIF1 α expression and activity. This mechanism may contribute to triggering an invasive phenotype in these cancer cells.

The data are novel and the experimental evidence is robust I have formulated below a few additional experiments in order to strengthen some of the conclusions and (or) to clarify the underlying phenotype upon MCU down regulation.

Specific remarks

1) In Fig 1A, the authors show a spectacular effect on mitochondrial Ca²⁺ uptake in H-Ras transformed malignant versus pre-malignant cells. A few points should be clarified here: (1) Is this effect due to an increase in MCU expression levels upon transformation; (2) Is this a general effect or could it be due to clonal differences as 2 clonal cell lines were compared? (3) The subsequent experiments were performed using siRNA silencing in 3 different metastatic TNBC models. Are the MCU expression levels equally high or increased in these cell lines as compared to non-metastatic cells? (4) Whereas the characterization of the effect of MCU down regulation was done using siRNA, the long term effects were obtained using stable shRNA down regulation. It is not shown that expression levels in the stable cell lines were equally efficiently down regulated as in acute siRNA experiments.

A western blot comparing MCU levels in these different cell lines would be informative to clarify these points.

2) The functional evaluation of MCU was performed by measuring acute agonist -induced calcium rises in the mitochondria. However, the effects of MCU silencing on the triggers of progression of metastasis, are long term effects and may be rather dependent on long-term changes in calcium levels. Are steady-state mitochondrial calcium levels also affected by MCU silencing? An assay of steady-state calcium in the mitochondria would exclude adaptive or compensatory corrections that could occur after MCU silencing, particularly in stable cell lines.

3) In Fig 2G, the authors make the point that MCU down regulation did not affect cell death in these cells. This experiment however was performed in conditions without apoptotic challenge. The number of viable cells was very high and obviously MCU knock down would not reveal a decrease in sensitivity to apoptosis in these conditions. What would be the effect of MCU silencing in conditions where calcium overload was triggered such as by staurosporine or ceramide? MCU silencing could render the cells less sensitive to conditions of increased apoptotic stress. This point would be relevant if the metastatic phenotype is characterized by increased MCU expression levels. The question then remains how such cells would be protected from calcium-induced cell death. Although these questions may be outside the scope of this paper, it would be informative to measure sensitivity to an apoptotic challenge after MCU down regulation.

4) An additional assay that could give more information about the MCU knock down conditions is

the occurrence of autophagy in these cells. This could be a mechanism that could compensate for at least part of the metabolic changes.

5) In Fig 3C, the authors should indicate what are the units used here for the NAD(P)H levels. It is clear that the life time (in ns) is indicative for the ratio, but it is not clear how the "levels" in Fig 3C were obtained and in which units they are expressed.

Referee #3 (Comments on Novelty/Model System):

Whereas this is a technically well-executed manuscript, the work largely focuses on in vitro experiments (with one exception) to claim that MCU-Ca²⁺ channel contributes to metastasis.

Metastasis is a complex multistep process and one that can only be tested in vivo. Any in vitro approach is limited to one aspect and cannot be used to conclude that will account for the whole. This is particularly important given that the only experiment in vivo showed a very strong effect of MCU-depletion in primary tumor growth. Thus, it is central to separate the contribution of the primary tumor to that of metastasis to support the author claims.

As per below, suggestions have been made to the authors to strength these aspects of the work.

Referee #3 (Remarks):

In this manuscript Tosatto et al suggest that Mitochondrial Calcium Uniporter (MCU) regulates cell migration via HIF-1a in triple negative breast cancer cells. Understanding mechanisms that support breast cancer progression and metastasis is of interest. In addition, recent findings indicate the potential role of mitochondrial genes and metabolism to cancer. Overall the experiments are technically well executed and the manuscript well written despite some technical and conceptual issues must be addressed to build on solid ground.

Overall, my biggest concern is novelty. My impression is that the manuscript represents an incremental rather than a substantial advance to the field as most of the observations were previously described and essentially the authors just aligned them altogether. In addition, the mechanistic basis of the finding remained elusive to the work and the functional and clinical validation of the contribution of MCU to metastasis is absent.

Major points.

1- The authors claim that MCU protein in TN BC is central to trigger pro migration and invasive functions. In particular they claim that MCU suppression strongly reduced TNBC migration potential, without affecting cell cycle or death. Interestingly they confirmed their finding by means of pharmacological inhibition of the MCU channel activity. Overall the authors use this cue as the central and most novel finding. In other words, the authors anchor their claims on the action of MCU to support pro metastatic function. However, a careful analysis of the data highlights that the main phenotypic function of MCU loss of function in vivo is reducing tumor growth.

In detail, MCU-depletion causes a 50% reduction in tumor growth in vivo, irrespectively of migration, invasion etc. This is an interesting observation but one that is established and is due to increase apoptosis (Curry et al 2013). Mitochondrial metabolism is essential to fuel cancer cells proliferation, thus the observation is not surprising. In addition, it is also well established that the tumor load is the main contributor to the risk of metastatic spread (tumor size in patients). Collectively, these observations imply that MCU main contribution to cancer is in supporting primary tumor growth and consequently MCU may also have an effect on metastatic burden. Thus, the latter could be easily attributed at the differences in primary tumor growth. This is a serious concern and largely hampered my enthusiasm with the manuscript.

Similarly, MCU Ca²⁺ mediated contribution to migration in vitro is not new (Tang et al 2015).

2- The current manuscript lacks of any clinical/physiological validation of the findings. Particularly, there is no experimental evidence related to metastatic functions. Indeed, given the observed effect in primary tumor, these analysis should be based on size-match lesions. As highlighted above, it is important to separate the contributions of MCU-mitochondria Ca^{2+} retention and metastasis compared to primary tumor effects. In vivo metastasis assays are a must to confirm the findings. Additionally, is the expression of MCU associated to tumor size, grade or any clinically pathological parameter including metastasis or time to metastasis? In size-match tumors, is MCU associated to increase risk of metastasis? Please note that metastasis and poor progression are different events so any misinterpretation throughout the text must be avoided.

3- In addition, the authors claim that MCU is central and exerts its function by means of controlling HIF-1a expression. If true, one would expect that tumors with high levels of MCU also express high levels of HIF-1a. In other words is there a significant correlation between MCU and HIF-1a in patient samples? Large expression data sets of TNBC samples are available, please confirm.

4- The authors initially build on the MCF10 model to describe the potential increase in MCU associated with metastasis. Unfortunately, the MCF10 model is valid as model of immortalized but not transform BC yet it has no relationship to TNBC. Please elaborate. Surprisingly, the model is not used at all in any of the subsequent studies were MCU contribution to cellular functions are validated. This is puzzling.

5- Finally, it is unclear how mechanistically MCU is upregulated in aggressive TNBC and not in premalignant cells. Similarly, it is unclear how HIF1a mediates the proposed contribution to migration and invasion beyond all the rest of potential well-described transcriptional targets and effectors. And finally, the authors described genes downstream of HIF-1a, however they must show that they account for a pro-metastatic phenotype in vivo and whose loss-of-function rescues MCU overexpression of redox stress contribution to metastasis.

Resubmission

27 January 2016

Response to referees' comments

Referee #1:

1. The effects of MCU silencing on cytosolic Ca^{2+} levels are ignored. MCU silencing was previously shown to decrease breast cancer cell migration by decreasing store-operated Ca^{2+} entry (SOCE) (Tang et al., 2015) an effect that had been proposed early on to reflect the Ca^{2+} buffering ability of mitochondria to relieve the Ca^{2+} -dependent inactivation of SOCE Ca^{2+} entry channels. Alteration in cytosolic Ca^{2+} signals, rather than alteration in mitochondrial matrix Ca^{2+} , might therefore contribute to the observed phenotype. To clarify this point, recordings of the cytosolic Ca^{2+} levels during stimulation with ATP, as well as quantification of the activity of SOCE channels in control, MCU silenced, and MCU overexpressing cells are required. In order to compare the mitochondrial Ca^{2+} response, the amount of Ca^{2+} stored in the ER should also be assessed by measuring the cytosolic Ca^{2+} response evoked by ATP in a Ca^{2+} -free medium, as well as the total amount of stored Ca^{2+} that can be mobilized by ionomycin.

Measurements of cytosolic $[Ca^{2+}]$ upon agonist stimulation, SOCE activity, and ER Ca^{2+} content are now reported in Figures S2 and S3. MCU silencing caused a decrease of agonist-induced cytosolic Ca^{2+} transients in BT-549 and MDA-MB-231 cell lines but not in MDA-MB-468 (Figure S2A). As mentioned in the revised manuscript, this difference could be ascribed to cell-type differences in the inhibitory effect that local high $[Ca^{2+}]$ microdomains play on Ins(1,4,5)P3R activity. Since MCU silencing reduces cell migration in all metastatic cell lines tested, this effect cannot be explained by MCU silencing effects on cytosolic Ca^{2+} transients.

Next, we measured SOCE activity in the three cell lines, as requested. In contrast to what reported by Tang et al., MCU silencing never caused a decrease in SOCE in our experimental conditions. Rather, an increase in SOCE was observed in MDA-MB-231 and MDA-MB-468

cell lines, in terms of both speed and maximal $[Ca^{2+}]$ entry and irrespective of the experimental protocol used to deplete Ca^{2+} store (either CPA, ionomycin or $Ins(1,4,5)P_3$ -coupled agonist) (Figure S2B-D). However, this effect was absent in BT-549 cells, again suggesting that the effects on migration do not depend on global cellular Ca^{2+} signaling. In addition, experiments in Ca^{2+} -free medium demonstrate that MCU silencing does not alter intracellular Ca^{2+} stores (Figure S2B-D). All together these data indicate that, in TNBC cells, the only cell line-independent effect of MCU silencing, that could explain the impairment in cell migration, is the reduction of mitochondrial Ca^{2+} uptake, as opposed to the other aspects of global Ca^{2+} homeostasis.

Finally, we analyzed Ca^{2+} signaling upon MCU overexpression. We observed an increase in mitochondrial Ca^{2+} uptake and a decrease in cytosolic $[Ca^{2+}]$ transients in the three cell lines (Figure S3A-B), which is in line with the buffering role that mitochondria can exert on cytosolic Ca^{2+} rises. MCU overexpression did not affect intracellular Ca^{2+} stores (Figure S3C-D) and, finally, the effect on SOCE was only marginal (Figure S3C-E).

2. The use of KB-R7943 as a "specific MCU inhibitor" (p. 5, last para) is questionable. This isothiourrea derivative is the prototypical inhibitor of the reverse mode of the plasma membrane Na^+ /Ca^{2+} exchanger (NCX) (Iwamoto et al., 2007; Iwamoto et al., 1996) and has been used extensively as such to explore the effects of reverse NCX inhibition. Besides this primary action, KB-R7943 was reported to inhibit TRPC channels (Kraft, 2007), L-type voltage-gated Ca^{2+} channels (Ouardouz et al., 2005), ryanodine receptors (Barrientos et al., 2009), NMDA receptors (Brustovetsky et al., 2011), and SOCE channels (Arakawa et al., 2000), while its action as an inhibitor of mitochondrial Ca^{2+} uptake has been questioned (Wiczler et al., 2014). Regardless of whether it inhibits the MCU or not, KB-R7943 is clearly a promiscuous pharmacologic agent with multiple Ca^{2+} -regulatory targets, and considering the long incubation time used here one cannot rely on such a tool as proof of implication of the MCU. If the authors intend to keep these pharmacological data they should at least validate the specificity of the inhibitor for mitochondrial Ca^{2+} uptake vs. other Ca^{2+} transporters by measuring its effects on cytosolic Ca^{2+} responses.

We agree with Referee's concern on the specificity of KB-R7943 treatment. In the revised manuscript, *in vitro* MCU silencing data are corroborated by *in vivo* tumorigenic experiments that rely on MCU^{-/-} cells (Figure 3). The specificity of these independent genetic approaches, together with the criticism raised by the referee, prompted us to remove from the revised manuscript the data based on KB-R7943 treatment.

3. Controls are lacking for the measurements of mitochondrial ROS production with Hyper and Hyper red (Fig 4). These probes are pH sensitive and parallel measurements of the matrix pH with dyes or fluorescent proteins are required to rule out that the responses do not reflect a matrix alkalinisation.

In the revised manuscript mitochondrial pH has been measured by means of the redox insensitive form of pHyper-dMito, i.e. SypHer2, demonstrating that MCU silencing does not affect matrix pH (Figure 5C).

4. To establish the causal link between MCU and Hif1a transcription, the authors should show that Hif1a transcription is restored by re-expressing the MCU protein in MCU-silenced cells.

In Figure S5 we now show that MCU re-expression in MCU silenced cells restores HIF1- α transcription to control values.

Other points

The quantification of the western blots shown in Fig. 5B are lacking error bars. Was this experiment performed only once?

The experiment was performed 5 times but, in the previous version of the manuscript, quantification referred only to the blot reported in the figure. In the revised version, quantification represents the average of 5 experiments (Figure 6B).

The original observation that initiated the study is the huge difference in mitochondrial Ca²⁺ uptake between premalignant and malignant cell lines (Fig 1A), but the genetic basis of this difference is not documented. What is the expression level of MCU in premalignant cell line? Is the difference in uptake due to altered MCU levels or to altered level of MCU regulatory proteins?

Related to the point above, the mechanism proposed would be strengthened by showing that the premalignant cell line with low mitochondrial Ca²⁺ uptake migrates poorly and produces low level of mitochondrial ROS.

We could not detect a direct correlation between protein levels of MCU complex components and agonist-induced mitochondrial Ca²⁺ uptake in premalignant versus malignant cell lines (not shown), indicating that the molecular mechanism underlying this difference is not directly related to this parameter, and other MCU mediated effects should be considered.

Nonetheless, a breast cancer gene expression dataset analysis demonstrated a positive correlation of MCU expression with tumor size and lymph node infiltration (Figure 1A-B). Interestingly, at the same time, MCUB, the dominant-negative isoform, negatively correlates with cancer parameters. We reasoned that these latter data would sustain our initial hypothesis even better. With this in mind, we decide to remove the data on the difference between premalignant and malignant cell lines.

Referee #2 (Remarks):

1) In Fig 1A, the authors show a spectacular effect on mitochondrial Ca²⁺ uptake in H-Ras transformed malignant versus pre-malignant cells. A few points should be clarified here: (1) Is this effect due to an increase in MCU expression levels upon transformation; (2) Is this a general effect or could it be due to clonal differences as 2 clonal cell lines were compared? (3) The subsequent experiments were performed using siRNA silencing in 3 different metastatic TNBC models. Are the MCU expression levels equally high or increased in these cell lines as compared to non-metastatic cells?

This referee raises important issues. Protein expression experiments indicate that the differences in mitochondrial Ca²⁺ uptake between malignant and pre-malignant cell lines is not due to increased MCU expression (not shown). As discussed in response to referee #1, we believe that the analysis of pre-malignant versus malignant cell lines in term of Ca²⁺ signaling requires further studies. However, in the revised manuscript, analysis of a breast cancer dataset demonstrate that MCU expression positively correlates with tumor size and lymph node infiltration (Figure 1A-B), while MCUB negatively correlates with these parameters. In the revised manuscript, we decided not to show the in vitro comparison of pre-malignant versus malignant cell lines, in favor of the more robust and significant breast cancer data.

(4) Whereas the characterization of the effect of MCU down regulation was done using siRNA, the long-term effects were obtained using stable shRNA down regulation. It is not shown that expression levels in the stable cell lines were equally efficiently down regulated as in acute siRNA experiments.

A western blot comparing MCU levels in these different cell lines would be informative to clarify these points.

In Figure S4A is now reported the strong downregulation of MCU protein levels in stable shMCU cells.

2) The functional evaluation of MCU was performed by measuring acute agonist -induced calcium rises in the mitochondria. However, the effects of MCU silencing on the triggers of progression of metastasis, are long term effects and may be rather dependent on long-term changes in calcium levels. Are steady-state mitochondrial calcium levels also affected by MCU silencing? An assay of steady-state calcium in the mitochondria would exclude adaptive or compensatory corrections that could occur after MCU silencing, particularly in stable cell lines.

Measurements of steady-state mitochondrial $[Ca^{2+}]$ in MCU knockdown and knockout cells demonstrate that long-term MCU silencing, in addition to lowering agonist-induced mitochondrial calcium uptake (Figures S4B, S4F), decreases resting mitochondrial $[Ca^{2+}]$ (Figures S4C, S4G). This effect was already observed in MCU^{-/-} mitochondria derived from skeletal muscle (Pan X. et al., Nature Cell Biology, 2013) and in isolated skeletal muscle fibers transfected with shMCU (Mammucari C. et al., Cell Reports, 2015). These results are expected, since compensatory mechanisms for depletion of MCU-mediated mitochondrial Ca^{2+} entry are not reported.

3) In Fig 2G, the authors make the point that MCU down regulation did not affect cell death in these cells. This experiment however was performed in conditions without apoptotic challenge. The number of viable cells was very high and obviously MCU knock down would not reveal a decrease in sensitivity to apoptosis in these conditions. What would be the effect of MCU silencing in conditions where calcium overload was triggered such as by staurosporine or ceramide? MCU silencing could render the cells less sensitive to conditions of increased apoptotic stress. This point would be relevant if the metastatic phenotype is characterized by increased MCU expression levels. The question then remains how such cells would be protected from calcium-induced cell death. Although these questions may be outside the scope of this paper, it would be informative to measure sensitivity to an apoptotic challenge after MCU down regulation.

Hall D.D. et al. reported that MCU knockdown does not cause significant effects on clonogenic cell survival of MDA-MB-231 cells exposed to irradiation or chemotherapeutic agents (Hall D.D. et al., PLOS ONE, 2014). In addition, the authors demonstrate that reduced mitochondrial Ca^{2+} uptake by expression of a dominant negative mutant of MCU does not affect ceramide-induced cell death. These results indicate that MCU is not involved in MDA-MB-231 cell survival.

4) An additional assay that could give more information about the MCU knock down conditions is the occurrence of autophagy in these cells. This could be a mechanism that could compensate for at least part of the metabolic changes.

In siMCU-transfected MDA-MB-231 cells LC3 protein levels (both LC3-I and LC3-II isoforms) were constantly decreased, indicating a possible effect on LC3 transcription. However, measurements of autophagy flux in siMCU cells treated with chloroquine (an inhibitor of autophagosome degradation), indicate that MCU silencing does not compromise autophagy activity. We report here a representative experiment.

5) In Fig 3C, the authors should indicate what are the units used here for the NAD(P)H levels. It is clear that the life time (in ns) is indicative for the ratio, but it is not clear how the "levels" in Fig 3C were obtained and in which units they are expressed.

The NADH/NADPH levels shown in Fig 3C (revised Fig 4C) are relative, normalised to the NADH levels in shControl cells. Using the equation in (Blacker et al., 2014), the relative NADH/NADPH ratios were calculated for each cell line from the lifetime values presented in Fig. 3B (revised Fig. 4B). Y axis and legend of figure 4B have been accordingly corrected.

Referee #3

1- The authors claim that MCU protein in TNBC is central to trigger pro migration and invasive functions. In particular they claim that MCU suppression strongly reduced TNBC migration potential, without affecting cell cycle or death. Interestingly they confirmed their finding by means of pharmacological inhibition of the MCU channel activity. Overall the authors use this cue as the central and most novel finding. In other words, the authors anchor their claims on the action of MCU to support pro metastatic function. However, a careful analysis of the data highlights that the main phenotypic function of MCU loss of function *in vivo* is reducing tumor growth.

In detail, MCU-depletion causes a 50% reduction in tumor growth in vivo, irrespectively of migration, invasion etc. This is an interesting observation but one that is established and is due to increase apoptosis (Curry et al 2013). Mitochondrial metabolism is essential to fuel cancer cells proliferation, thus the observation is not surprising. In addition, it is also well established that the tumor load is the main contributor to the risk of metastatic spread (tumor size in patients). Collectively, these observations imply that MCU main contribution to cancer is in supporting primary tumor growth and consequently MCU may also have an effect on metastatic burden. Thus, the latter could be easily attributed at the differences in primary tumor growth. This is a serious concern and largely hampered my enthusiasm with the manuscript.

Similarly, MCU Ca²⁺ mediated contribution to migration in vitro is not new (Tang et al 2015).

To address the concerns raised by this referee we performed additional *in vivo* experiments that conclusively demonstrate that MCU depletion reduces a) tumor growth and b) metastasis formation independently of primary tumor size. We took advantage of the CRISPR/Cas9 technology to develop two independent MCU^{-/-} MDA-MB-231 clones that were injected into the fat pad of SCID mice. MCU knockout markedly reduces tumor growth (Figure 3A), confirming our previous observation with shMCU xenografts (replaced in full by the MCU^{-/-} data). In addition, data on size-matched tumors demonstrate that MCU depletion reduces lymph node infiltration and lung metastasis independently on primary tumor growth (Figures 3B-E).

We also demonstrate that MCU silencing does not affect cell death (Figure 2D), which is in line with Curry et al. 2013, where it was shown that siMCU does not affect MDA-MB-231 cell viability per se, while it is able to potentiate ionomycin-induced death.

Thus MCU has a dual effect on tumor progression, both on primary tumor size and on metastasis, in conditions in which cell viability is unaffected.

2- The current manuscript lacks of any clinical/physiological validation of the findings. Particularly, there is no experimental evidence related to metastatic functions. Indeed, given the observed effect in primary tumor, these analysis should be based on size-match lesions. As highlighted above, it is important to separate the contributions of MCU-mitochondria Ca^{2+} retention and metastasis compared to primary tumor effects. In vivo metastasis assays are a must to confirm the findings. Additionally, is the expression of MCU associated to tumor size, grade or any clinically pathological parameter including metastasis or time to metastasis? In size-match tumors, is MCU associated to increase risk of metastasis? Please note that metastasis and poor progression are different events so any misinterpretation throughout the text must be avoided.

Part of the answer is reported in response to question #1 of this same referee.

Concerning the association of MCU expression with clinical parameters, dataset analysis of breast tumors demonstrate that MCU levels positively correlates with tumor size and lymph node infiltration (Figure 1A-B). The data is further supported by the negative correlation of these parameters with MCUB, the dominant-negative isoform of the channel.

Finally, the text has been revised to avoid any confusion between metastasis and poor progression.

3- In addition, the authors claim that MCU is central and exerts its function by means of controlling HIF-1a expression. If true, one would expect that tumors with high levels of MCU also express high levels of HIF-1a. In other words is there a significant correlation between MCU and HIF-1a in patient samples? Large expression data sets of TNBC samples are available, please confirm.

Dataset analysis of breast cancer samples demonstrates a positive correlation between MCU expression and HIF-1a (Figure 6L). This data is further supported by the positive correlation of MCU expression with HIF-1a-regulated genes (Figure 6M).

4- The authors initially build on the MCF10 model to describe the potential increase in MCU associated with metastasis. Unfortunately, the MCF10 model is valid as model of immortalized but not transform BC yet it has no relationship to TNBC. Please elaborate. Surprisingly, the model is not used at all in any of the subsequent studies were MCU contribution to cellular functions are validated. This is puzzling.

We agree that the analysis of pre-malignant versus malignant cell lines in term of Ca^{2+} signaling requires further studies, starting from a better choice of cell lines.

In the revised manuscript, analysis of a breast cancer dataset demonstrates that MCU expression positively correlates with tumor size and lymph node infiltration while MCUB negatively correlates with these parameters (Figure 1A-B). We thus decided not to show the *in vitro* comparison of mitochondrial Ca^{2+} uptake, in favor of the more robust and significant breast cancer samples data.

5- Finally, it is unclear how mechanistically MCU is upregulated in aggressive TNBC and not in premalignant cells. Similarly, it is unclear how HIF1a mediates the proposed contribution to migration and invasion beyond all the rest of potential well-described transcriptional targets and effectors. And finally, the authors described genes downstream of HIF-1a, however they must show

that they account for a pro-metastatic phenotype *in vivo* and whose loss-of-function rescues MCU overexpression of redox stress contribution to metastasis.

The increase in mitochondrial Ca^{2+} uptake in TNBC is likely due to an increase of MCU activity, which not necessarily relies on MCU protein expression, but could be due to regulation of the channel activity by MICUs and EMRE. Post-transcriptional modifications of channel components and regulators may also take place. However, as discussed above, we decided not to include these data in the revised manuscript in order to better investigate this issue.

In the new manuscript, in addition to the previous observation that HIF1a mediates MCU-dependent migration (Figure 6K), breast cancer dataset analysis demonstrates that MCU expression correlates with HIF1a and HIF1a regulated genes (Figure 6L-M).

The HIF1a regulated genes analyzed in this manuscript have already been reported to regulate migration and invasion *in vitro* and *in vivo*, as summarized hereafter. Nonetheless, we confirmed the effect of their downregulation on migration in MDA-MB-231 cells, as reported in the figure below.

a) Pharmacological inhibition of HK by Clotrimazole treatment inhibits migration of MDA-MB-231 cells (Furtado C.M. et al., PLOS ONE, 2012). In addition the α -tocopherol derivative ESeroS-GS downregulates the expression of HK II and other metabolic enzymes leading to a decrease in oxidative phosphorylation and glycolysis in MDA-MB-231 cells. Sub-toxic concentration of ESeroS-GS treatment results in the impairment of F-actin cytoskeleton assembly and the consequently migratory and invasive ability of MDA-MB-231 cells. (Zhao L. et al., European Journal of Pharmacology, 2014);

b) PGI/AMF (G6PI) is a key gene to MET in the later stage of metastasis during breast cancer progression. Silencing of PGI/AMF expression in MDA-MB-231 cells induced MET. (Tatsuyoshi Funasaka, Victor Hogan, and Avraham Raz Phosphoglucose Isomerase/Autocrine Motility Factor Mediates Epithelial and Mesenchymal Phenotype Conversions in Breast Cancer Cancer Res 2009; 69: (13). July 1, 2009). In addition, silencing of PGI/AMF expression in MDA-MB-231 cells led to decreased motility, and invasiveness and suppressed pulmonary metastases of MDA-MB-231 cells *in vivo*. (Ahmad A. et al., Cancer Res, 2011);

c) LOX regulates cell migration and cell-matrix adhesion formation. LOX expression is up-regulated in metastasis from breast cancer. Treatment of MDA-MB-231 with B-aminopropionitrile (BAPN), an irreversible inhibitor of LOX catalytic activity, leads to a

significant decrease in cell motility/migration and adhesion formation. (Payne S.L. et al., Cancer Res 2005);

d) Pyruvate Dehydrogenase Kinase 1 (PDK1) is a negative regulator of pyruvate dehydrogenase (PDH), the enzyme that catalyze the conversion of pyruvate to acetyl-CoA and thus allows the entry of pyruvate into the TCA cycle.

PDK1 expression is upregulated in many cancers due to increased HIF activity and contributes to the shift from oxidative to glycolytic metabolism. The subsequent increase in lactate leads to increased invasion and migration.

e) Carbolic anhydrase inhibitors are effective at inhibiting tumor cell growth in vitro and in vivo. (Neri, D. et al., Nat. Rev. Drug Discovery 2011). In particular sulfonamide-based inhibitors, which are highly selective for CA IX, selectively inhibit cell migration and spreading in hypoxia. One of this compounds (S4) effectively inhibits the spontaneous metastasis formation in MDA-MB-231 xenografts (Gieling R.G. et al., Journal of Medicinal Chemistry, 2012).

2nd Editorial Decision

15 February 2016

Thank you for the re-submission of your manuscript to EMBO Molecular Medicine. We have now received the enclosed reports from the referees that were asked to re-assess it. As you will see the reviewers are now globally supportive and I am pleased to inform you that we will be able to accept your manuscript pending the following final amendments:

Please address the minor comments from referees 1 and 3 by adding extra-discussion as requested. Furthermore, you will see that referee 3 suggest to perform an additional control. Should you already have data pertaining to this issue or can perform this experiment easily and in a timely manner, I would encourage you to do so. However, we do not qualify this as mandatory for acceptance. Please provide a letter including the reviewer's reports and your detailed responses to their comments (as Word file).

Please submit your revised manuscript within two weeks. I look forward to seeing a revised form of your manuscript as soon as possible.

***** Reviewer's comments *****

Referee #1 (Comments on Novelty/Model System):

In my opinion the authors have to a very large extent addressed the comments and concern of the 3 referees. In particular they have removed the data about differences in mitochondrial calcium uptake between premalignant and malignant cell line, for which they had no clear molecular explanation. In this revised version they have instead used an analysis of a breast cancer gene expression dataset. Based on this analysis they find a clear positive correlation of MCU expression with tumor size and lymph node infiltration. In addition there was a negative correlation with the dominant negative MCUb. These new data are indeed a much better base supporting their working hypothesis.

The experimental data in this study are very robust and clearly show a contribution of mitochondrial calcium uptake (mediated by MCU) to metastasis.

In particular, they showed that MCU depletion reduced cell growth and invading capacity in in vitro experiments, and confirmed this in vivo by showing effects on primary tumor growth, lymph node infiltration and lung metastasis formation.

Concerning the cellular events that underlie this process, the authors admit that not all parameters are resolved and they may involve a complex rearrangement of mitochondrial and cellular metabolism. Nevertheless they could provide good evidence that mROS production was

significantly blunted upon MCU silencing, although this may not be the only factor involved. Downstream of mROS the authors found a regulatory mechanism based on downregulation of HIF-1 transcription. A detailed analysis of HIF-1 and its target genes, showed that HIF-1 may represent the key effector of the siMCU-mediated phenotype, and that the positive correlation between MCU expression and HIF-1 and its target genes exists in human breast cancer samples.

As the authors have sufficiently answered the concerns of the referees and the data are novel and important, I believe that this paper should be accepted for publication.

Referee #1 (Remarks):

The authors have demonstrated in a convincing way that the expression of the MCU and the associated mitochondrial calcium uptake, are correlated with tumor size and lymph node infiltration. MCU down-regulation hampered cell motility and invasiveness and reduced tumor growth, lymph node infiltration and lung metastasis in xenografts.

The study also showed that the phenotype of MCU-silenced cells is characterized by reduced ROS production and reduced expression of HIF-1 and its targets genes.

The data are robust and convincing and suggest that MCU plays a central role in metastasis and may become a novel therapeutic target.

The revised version is significantly amended and is very clearly written.

I have only a minor remark concerning the effect of MCU silencing on the ability of TNBC cells to invade a collagen matrix (Fig2). There was in addition an effect on cell growth after 7 days as revealed by a colony formation assay.

Is this observation not conflicting with the data in Fig 1I-K, where there was no effect on proliferation?

Moreover, the authors eliminated apoptosis and cell cycle arrest as possible causes. It is not clear however what the authors suggest as the underlying reason for this inhibition of cell growth.

Similarly, the in vivo data on SCID mice also show an effect on tumor growth (Fig 3A) in addition to the effects on infiltration and metastasis (Fig 3B-E). It is not clear from the discussion if the authors suggest that both effects are connected to the same signaling mechanism or if additional pathways may be involved to explain the effect on tumor growth.

On p9, line 12-13 the sentence may be wrongly formulated as it states that: "molecular knock down of mitochondrial calcium signaling is NEEDED for rapid tumor progression and metastasis formation", whereas in fact MCU knock-down was shown to IMPAIR these processes.

Referee #2 (Comments on Novelty/Model System):

After a revised version of the manuscript was reviewed, this reviewer is largely satisfied with the two main actions taken by the authors to address the main concerns. This included the thorough analysis of clinical samples from large publicly available data sets to provide clinical framework for the findings. In addition, the new experiments performed in metastasis mouse models are reasonable and fit with the technical standards of the field.

Referee #2 (Remarks):

The authors have taken a serious endeavor to turn around the manuscript compared with the previously submitted version. This reviewer feels satisfied with the experimental approaches taken including but not limited to the tumor-size matched metastasis experiments and the clinical relevance of the findings. These were pivotal to sustain an otherwise interesting mechanistic study but limited to the cell-culture aficionados. In addition, all text changes and efforts to streamline and limit the manuscript to cancer avoiding premalignant role has a significant impact on the flow and the relevance.

The explanations and data provided for the rest of the points are reasonable and support their case.

Overall, the manuscript has largely improved through this round of revision and is now a better fit for the journal.

Referee #3 (Comments on Novelty/Model System):

That MCU levels impacts cancer cells migration by controlling ROS levels and HIF1a signalling is new and important. The results were obtained both in vitro and in vivo using appropriate animal and cellular models.

Referee #3 (Remarks):

The authors have performed extensive experimental work. The new body of evidence reinforces the initial findings but also generates some confusion that needs to be addressed. CRISP/CAS9 is now used instead of pharmacology to show that MCU depletion decreases the in vivo tumorigenic capacity of breast cancer cells in mice xenografts (Fig 3). This provides much more solid evidence for a causal role of the MCU than the previous pharmacological approach. However, the impact of MCU deletion on cytosolic Ca²⁺ signals has not been tested and the authors do not conclusively rule out a role for altered cytosolic Ca²⁺ signalling. Cytosolic Ca²⁺ recordings were performed in shMCU-silenced cells as requested and revealed complex and cell-line dependent effects of MCU depletion (Fig. S2, S3). MCU silencing reduced Ca²⁺ release from intracellular stores in two out of 3 cell lines tested while the third cell line had excessive store-operated Ca²⁺ entry. The authors consider these effects as irrelevant, because the phenotype is not consistent for the three cell lines. I would argue that there is a consistent phenotype here, which is that the cytosolic Ca²⁺ signals are altered in every cell line tested. Even if the defect differ in nature, the cytosolic Ca²⁺ homeostasis is perturbed every time that the MCU is silenced, and this could account for the in vitro phenotype of these cells. Whether a similar Ca²⁺ defects could account for reduced progression of grafted cells in vivo is not known, because only mitochondrial Ca²⁺ responses are shown for the MCU null clones (Fig. S4E).

To conclude that "impairment in cell migration can be explained only by the specific reduction in mitochondrial Ca²⁺ uptake." (p. 7, results, end of first para) or that "the only cell line-independent effect of MCU silencing that could explain the impairment in cell migration is the reduction of mitochondrial Ca²⁺ uptake" as the authors do in their rebuttal letter is both an overstatement and a misjudgement. Ca²⁺ signals are encoded in space, time, and frequency, and Ca²⁺-dependent effector functions are not directly proportional to the amplitude of Ca²⁺ elevations. The "inconsistent" Ca²⁺ defects could both alter cell migration if, for instance, excessive Ca²⁺ signals at the plasma membrane interfere with integrin signalling while reduced release from stores hinders actin remodelling. To ignore these findings completely in the discussion and retain only the mitochondrial defect as cause looks like cherry picking. These findings deserve a transparent evaluation and discussion. Cytosolic Ca²⁺ responses of MCU null cells should be included to allow a proper evaluation of the effects of MCU deletion on cellular Ca²⁺ homeostasis.

Referee #1 (Remarks):

I have only a minor remark concerning the effect of MCU silencing on the ability of TNBC cells to invade a collagen matrix (Fig2). There was in addition an effect on cell growth after 7 days as revealed by a colony formation assay.

Is this observation not conflicting with the data in Fig 1I-K, where there was no effect on proliferation?

Moreover, the authors eliminated apoptosis and cell cycle arrest as possible causes. It is not clear however what the authors suggest as the underlying reason for this inhibition of cell growth.

Similarly, the in vivo data on SCID mice also show an effect on tumor growth (Fig 3A) in addition to the effects on infiltration and metastasis (Fig 3B-E). It is not clear from the discussion if the authors suggest that both effects are connected to the same signaling mechanism or if additional pathways may be involved to explain the effect on tumor growth.

The reviewer points out an important issue that deserves further clarification.

Concerning the comparison between data on cell proliferation (Fig 1) and on colony formation and invasion (Fig 2), we believe that a fundamental difference is the time point of the analysis. Cell proliferation analyses reported in Figure 1 are short-term experiments. Our data demonstrate that 72 hours are not enough time for a hypothetical effect of MCU silencing on cell number. In the context of Figure 1, this data validates the 3-days migration experiment, since differences in cell numbers are excluded. However, when analysis of cell growth is extended to 7 days, differences emerge, and the data is supported by the impairment of the in vivo tumor growth. We hypothesise that the mechanism underlying long-term growth impairment could be a subtle slow-down in cell growth that has not been unmasked by our cell cycle analysis. Whether the effect of MCU silencing on cell growth involves the same signalling pathways regulated during metastasis control is unknown. Surely, these issues deserve further investigation.

On p9, line 12-13 the sentence may be wrongly formulated as it states that: "molecular knock down of mitochondrial calcium signaling is NEEDED for rapid tumor progression and metastasis formation", whereas in fact MCU knock-down was shown to IMPAIR these processes.

We are grateful to the reviewer for this observation. The text has been corrected.

Referee #3 (Remarks):

...the impact of MCU deletion on cytosolic Ca²⁺ signals has not been tested and the authors do not conclusively rule out a role for altered cytosolic Ca²⁺ signalling. Cytosolic Ca²⁺ recordings were performed in shMCU-silenced cells as requested and revealed complex and cell-line dependent effects of MCU depletion (Fig. S2, S3). MCU silencing reduced Ca²⁺ release from intracellular stores in two out of 3 cell lines tested while the third cell line had excessive store-operated Ca²⁺ entry. The authors consider these effects as irrelevant, because the phenotype is not consistent for the three cell lines. I would argue that there is a consistent phenotype here, which is that the cytosolic Ca²⁺ signals are altered in every cell line tested. Even if the defect differ in nature, the cytosolic Ca²⁺ homeostasis is perturbed every time that the MCU is silenced, and this could account for the in vitro phenotype of these cells. Whether a similar Ca²⁺ defects could account for reduced progression of grafted cells in vivo is not known, because only mitochondrial Ca²⁺ responses are shown for the MCU null clones (Fig. S4E).

To conclude that "impairment in cell migration can be explained only by the specific reduction in mitochondrial Ca²⁺ uptake." (p. 7, results, end of first para) or that "the only cell line-independent effect of MCU silencing that could explain the impairment in cell migration is the reduction of mitochondrial Ca²⁺ uptake" as the authors do in their rebuttal letter is both an overstatement and a misjudgement. Ca²⁺ signals are encoded in space, time, and frequency, and Ca²⁺-dependent effector functions are not directly proportional to the amplitude of Ca²⁺ elevations. The "inconsistent" Ca²⁺ defects could both alter cell migration if, for instance, excessive Ca²⁺ signals at the plasma membrane interfere with integrin signalling while reduced release from stores hinders actin remodelling. To ignore these findings completely in the discussion and retain only the mitochondrial defect as cause looks like cherry picking. These findings deserve a transparent evaluation and discussion.

Cytosolic Ca²⁺ responses of MCU null cells should be included to allow a proper evaluation of the effects of MCU deletion on cellular Ca²⁺ homeostasis.

We thank the reviewer for these insightful comments. We have now included the data that demonstrates that MCU deletion does not alter cytosolic Ca²⁺ transients in CRISP9/Cas clones (Appendix Figure S4H). We believe that this data excludes the possible, and certainly intriguing, role of altered cytosolic Ca²⁺ transients in metastasis proposed by the reviewer. Nonetheless, the sentence on page 7 stating "Thus, the impairment in cell migration triggered by MCU silencing *can be explained only by* the specific reduction in mitochondrial Ca²⁺ uptake..." has been changed to "Thus, the impairment in cell migration triggered by MCU silencing *is most likely due to* the specific reduction in mitochondrial Ca²⁺ uptake..." .

Discussion on this topic is now present at page 14.

Corresponding Author Name: Cristina Mammucari

Manuscript Number: EMM-2016-06255